# A-type FHFs mediate resurgent currents through TTX-resistant voltage-gated sodium channels

Yucheng Xiao[1]*, Jonathan W Theile[2], Agnes Zybura[3], Yanling Pan[1], Zhixin Lin[2], Theodore R Cummins[1]*

[1]Biology department, School of Science, Indiana University Purdue University Indianapolis, Indianapolis, United States; [2]Icagen LLC, 4222 Emperor Blvd #350, Durham, United States; [3]Program in Medical Neuroscience, Paul and Carole Stark Neurosciences Research Institute, Indiana University School of Medicine, Indianapolis, United States

**Abstract** Resurgent currents ($I_{NaR}$) produced by voltage-gated sodium channels are required for many neurons to maintain high-frequency firing and contribute to neuronal hyperexcitability and disease pathophysiology. Here, we show, for the first time, that $I_{NaR}$ can be reconstituted in a heterologous system by coexpression of sodium channel α-subunits and A-type fibroblast growth factor homologous factors (FHFs). Specifically, A-type FHFs induces $I_{NaR}$ from Nav1.8, Nav1.9 tetrodotoxin (TTX)-resistant neuronal channels, and, to a lesser extent, neuronal Nav1.7 and cardiac Nav1.5 channels. Moreover, we identified the N-terminus of FHF as the critical molecule responsible for A-type FHFs-mediated $I_{NaR}$. Among the FHFs, FHF4A is the most important isoform for mediating Nav1.8 and Nav1.9 $I_{NaR}$. In nociceptive sensory neurons, FHF4A knockdown significantly reduces $I_{NaR}$ amplitude and the percentage of neurons that generate $I_{NaR}$, substantially suppressing excitability. Thus, our work reveals a novel molecular mechanism underlying TTX-resistant $I_{NaR}$ generation and provides important potential targets for pain treatment.

*For correspondence:
yuchxiao@indiana.edu (YX);
trcummin@iu.edu (TRC)

## Editor's evaluation

This is an exciting and important study that constitutes a major advance in the molecular understanding of resurgent Na current. Reproducing resurgent current by expression of two proteins has never been done: here, the authors have for the first time molecularly reconstituted Na channels that produce resurgent Na current. Not only do these experiments satisfactorily and convincingly address a long-standing question in the field, but they also open the door to molecular manipulation of this current, potentially of significant practical use given the proposed role of the current in several disorders and disease states, including pain. The work will be of interest to many neuroscientists.

## Introduction

Voltage-gated sodium channels (VGSCs) are crucial determinants of action potentials in almost all excitable tissues. VGSCs are composed of a functional pore-forming α-subunit associated with auxiliary β-subunits (*Catterall et al., 2005*). VGSCs also interact with other intracellular proteins, such as fibroblast growth factor homologous factors (FHFs) and calmodulin (*Catterall et al., 2005*; *Wildburger et al., 2015*). Although the α-subunit is sufficient to produce a functional VGSC, interacting partners can influence multiple properties of the α-subunits, regulating neuronal excitability (*Namadurai et al.,*

*2015*). One of the most striking influences is generation of resurgent sodium currents ($I_{NaR}$) (*Lewis and Raman, 2014*).

$I_{NaR}$ were originally observed in cerebellar Purkinje neurons (*Raman and Bean, 1997*) and have been identified in cerebellum, brainstem, trigeminal ganglia, and dorsal root ganglion (DRG) neurons (*Afshari et al., 2004*; *Enomoto et al., 2006*; *Kim et al., 2010*). $I_{NaR}$ can enhance high-frequency firing in many neurons (*Raman and Bean, 1997*; *Xie et al., 2016*), and aberrant $I_{NaR}$ have been implicated in multiple human diseases including pain disorders (*Jarecki et al., 2010*; *Patel et al., 2016*; *Theile et al., 2011*; *Tanaka et al., 2016*). Unlike classic sodium currents that are activated by step depolarizations, $I_{NaR}$ are atypical sodium currents evoked by step repolarizations. Navβ4 has been implicated as a major contributor to $I_{NaR}$ generation (*Grieco et al., 2005*; *Barbosa et al., 2015*; *Cannon and Bean, 2010*). The most direct evidence supporting the Navβ4 mechanism is that a short peptide derived from the C-terminal tail of Navβ4 can reconstitute the $I_{NaR}$. However, the Navβ4 mechanism remains controversial for at least two reasons: (1) Navβ4 knockout or knockdown does not abolish $I_{NaR}$ in central (*White et al., 2019*; *Ransdell et al., 2017*) or peripheral neurons (*Xiao et al., 2019*), but rather results in only partial to no reduction of $I_{NaR}$; and (2) importantly, coexpression of full-length Navβ4 with VGSC α-subunits fails to reconstitute $I_{NaR}$ in heterologous systems. Therefore, other molecular mechanisms for $I_{NaR}$ generation remain to be uncovered.

FHFs are widely distributed throughout the central nerve system (CNS)/peripheral nerve system (PNS). They represent an important group of auxiliary VGSC subunits that influence neuronal excitability. FHF is a subfamily of the fibroblast growth factor (FGF) superfamily. They can bind to the VGSC C-terminal tails and can modulate VGSCs functional properties, trafficking, and axonal localization (*Liu et al., 2001*; *Goetz et al., 2009*; *Wittmack et al., 2004*; *Lou et al., 2005*; *Wang et al., 2011b*). There are two main types of FHFs: A-type and B-type. The former has four isoforms (FHF1A [or FGF12-1a], FHF2A [or FGF13-1a], FHF3A [or FGF11-1a], FHF4A [or FGF14-1a]). There is emerging evidence that FHFs regulate $I_{NaR}$ generation in neurons. In DRG neurons, overexpression of FHF2A and FHF2B (also known as FGF13-1b) decreases and increases Nav1.6 $I_{NaR}$, respectively (*Barbosa et al., 2017*). In contrast, FHF4A, which has high sequence similarity to FHF2A, has been proposed to directly mediate $I_{NaR}$ generation by Nav1.6. FHF4 knockout significantly reduced $I_{NaR}$ in Purkinje neurons, which is mainly carried by Nav1.6, and a peptide corresponding to FHF4A residues 50–63 induced robust $I_{NaR}$ in CA3 neurons (*White et al., 2019*). However, both FHF2A and FHF4A have been shown to induce accumulation of rapid-onset long-term inactivation when coexpressed with Nav1.6 in heterologous systems (*Venkatesan et al., 2014*; *Dover et al., 2010*). In addition, the reduction in cerebellar Purkinje neuron $I_{NaR}$ with FHF4 knockdown has been proposed to be due to an indirect effect involving FHF4B modulation of channel inactivation (*Yan et al., 2014*). Therefore, there is a lack of compelling evidence supporting a specific molecular mechanism of $I_{NaR}$ generation.

In this study, we report that A-type FHFs directly mediate resurgent sodium current generation in Nav1.8 and Nav1.9 sensory neuron VGSCs, and show for the first time that $I_{NaR}$ can be reconstituted in a heterologous system (ND7/23 and HEK293 cells, respectively) by coexpressing full-length A-type FHFs with VGSC α-subunits. These FHF-mediated $I_{NaR}$ are independent of Navβ4. The novel FHF-mediated $I_{NaR}$ could be fully reproduced by the amino acids 2–21 from the A-type FHF N-terminus. We also show that while FHF2A could induce small $I_{NaR}$ with Nav1.5 and Nav1.7, FHF4A did not induce Nav1.5, Nav1.6, or Nav1.7 $I_{NaR}$ in heterologous expression systems. We further show that reduction of FHF4A-mediated tetrodotoxin (TTX)-resistant $I_{NaR}$ substantially downregulated excitability of nociceptive DRG neurons. Because Nav1.7–Nav1.9 are predominantly expressed in neurons of DRG and trigeminal ganglia, and are crucial for pain perception and transmission (*Cummins et al., 2004*; *Cox et al., 2006*; *Huang et al., 2014*; *Dib-Hajj et al., 2015*; *Huang et al., 2013*; *Cummins et al., 2007*; *Dib-Hajj et al., 2010*), our work not only uncovers a novel mechanism of $I_{NaR}$ generation in sensory neurons, but also identifies an exciting target for the development of new pain treatments.

## Results

### A-type FHFs mediate $I_{NaR}$ in heterologously expressed Nav1.8 and Nav1.9

A-type FHFs can modulate TTX-sensitive VGSC inactivation and $I_{NaR}$ (*White et al., 2019*; *Barbosa et al., 2017*; *Yan et al., 2014*); however, it is unknown if A-type FHFs impact the functional properties

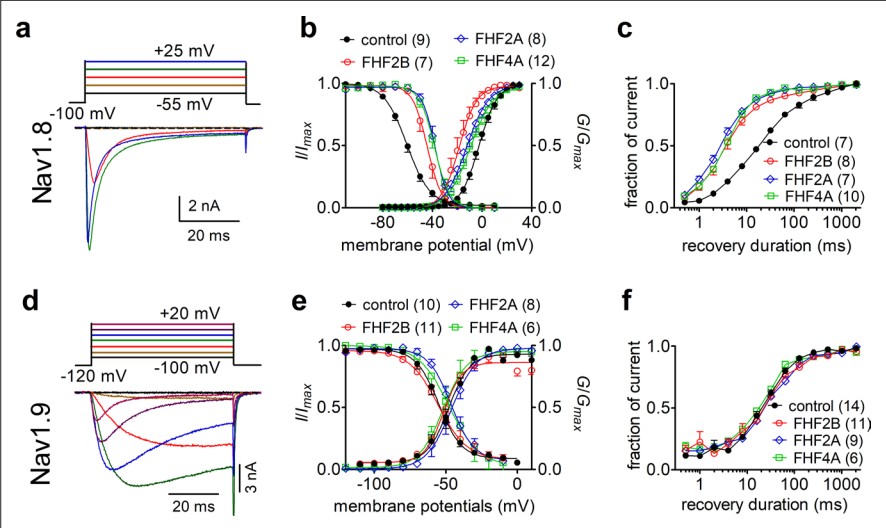

**Figure 1.** Fibroblast growth factor homologous factors (FHFs) differentially modulated the gating properties of Nav1.8 and Nav1.9 in heterologous systems. (**a**) Family of classical currents recorded from ND7/23 cells expressing recombinant Nav1.8. Currents were elicited by 50 ms depolarizing voltage steps from +25 mV to −55 mV in −10 mV increments from a holding potential of −100 mV (inset). (**b**) Effects of FHF2B, FHF2A, and FHF4A on steady-state activation (p<0.0001, 0.0035, 0.0077 vs. control, respectively) and inactivation (p=0.0002, <0.0001, <0.0001 vs. control, respectively) of Nav1.8. (**c**) FHF2B, FHF2A, and FHF4A accelerated the recovery rate from Nav1.8 inactivation. The time constants estimated from single-exponential fits were 29.71 ± 2.54 ms (control), 5.81 ± 1.03 ms (FHF2B, p<0.0001 vs. control), 4.45 ± 0.43 ms (FHF2A, p<0.0001 vs. control), and 5.46 ± 0.40 ms (FHF4A, p<0.0001 vs. control). (**d**) Family of classical currents recorded from HEK293 cells expressing recombinant Nav1.9. Currents were elicited by 50 ms depolarizing voltage steps from +20 mV to −100 mV in −20 mV increments from a holding potential of −120 mV (inset). (**e**) Effects of FHF2B, FHF2A, and FHF4A on steady-state activation (p=0.1832, 0.0171, 0.3215 vs. control, respectively) and inactivation (p=0.175, 0.5978, 0.636 vs. control, respectively) of Nav1.9. (**f**), FHF2B, FHF2A, and FHF4A did not affect the recovery rate from Nav1.9 inactivation. The time constants estimated from single-exponential fits were 38.46 ± 4.64 ms (control), 48.99 ± 6.93 ms (FHF2B, p=0.2041 vs. control), 49.72 ± 6.81 ms (FHF2A, p=0.1745 vs. control), and 31.95 ± 2.84 ms (FHF4A, p=0.4786 vs. control). In (**a–c**), cells were pretreated with 500 nM TTX. In (**c, f**), recovery from inactivation was assayed by the protocol that the cells were prepulsed to 0 mV for 50 ms to inactivate sodium channels and then brought back to −100 mV for increasing recovery durations before the test pulse to 0 mV. Filled circles, open circles, open diamond, and open squares represent control, FHF2B, FHF2A, and FHF4A, respectively. The number of separate cells tested is indicated in parentheses. Data points are shown as mean ± SE. The $V_{1/2}$ values for activation and inactivation are summarized in *Table 1*.

of the TTX-resistant sodium channels Nav1.8 and Nav1.9. Therefore, we first asked whether FHF2A and FHF4A, which are widely expressed in DRG neurons, modulate sodium currents in cells expressing recombinant Nav1.8 and Nav1.9. As previously shown in ND7/23 and HEK293 heterologous cell expression systems (*Xiao et al., 2019*; *Lin et al., 2016*), Nav1.8 generated a slow-inactivating TTX-resistant current, while Nav1.9 produced an ultra-slow-inactivating TTX-resistant current that activated at hyperpolarized potentials (*Figure 1a and d*). Although ND7/23 are from a rat DRG/mouse N18Tg2 neuroblastoma hybridoma cell line, they do not express endogenous Nav1.8 currents (*John et al., 2004*; *Lee et al., 2019*) and are used here as they typically express recombinant Nav1.8 currents at higher levels than HEK293 cells. Here, we show that FHF2A, FHF2B, and FHF4A, when coexpressed with Nav1.8, shifted the voltage dependence of activation by >7 mV in the negative direction and shifted the voltage dependence of steady-state inactivation by >15 mV in the positive direction (*Figure 1b*, *Table 1*). FHF2A, FHF2B, and FHF4A also accelerated recovery rate from inactivation of Nav1.8 (*Figure 1c*). When coexpressed with Nav1.9, FHF2A and FHF4A, but not FHF2B, positively shifted the voltage dependence of steady-state inactivation by ~10 mV. Distinct from Nav1.8, none of the three FHF isoforms altered the voltage dependence of activation or rate for recovery from inactivation of Nav1.9 (*Figure 1e and f*, *Table 1*), suggesting that FHFs differentially regulate TTX-resistant VGSCs.

**Table 1.** Gating properties of Nav1.8 and Nav1.9 in the presence of fibroblast growth factor homologous factors (FHFs).

Midpoint voltages of the steady-state activation and inactivation curves in *Figure 1* were determined with a standard Boltzmann distribution fit. *$p<0.05$ and @$p<0.001$ vs. respective control condition. The number of separate cells tested is indicated in parentheses. Note that the liquid junction potential for these solutions was <8 mV; data were not corrected to account for this offset.

| Construct | V1/2 (mV) | Control | FHF2A | FHF2B | FHF4A |
|---|---|---|---|---|---|
| Nav1.8 | Activation | −2.4 ± 1.5 (9) | −12.4 ± 2.7@ (8) | −18.5 ± 2.7@ (7) | −9.0 ± 1.9* (12) |
| | Inactivation | −59.6 ± 1.8 (9) | −37.9 ± 1.9@ (8) | −44.3 ± 2.3@ (7) | −39.0 ± 2.2@ (12) |
| Nav1.9 | Activation | −47.8 ± 2.5 (10) | −46.0 ± 1.2 (8) | −50.9 ± 2.0 (11) | −50.5 ± 4.8 (6) |
| | Inactivation | −55.2 ± 2.3 (10) | −45.9 ± 2.5* (8) | −53.2 ± 4.2 (11) | −45.2 ± 4.3 (6) |

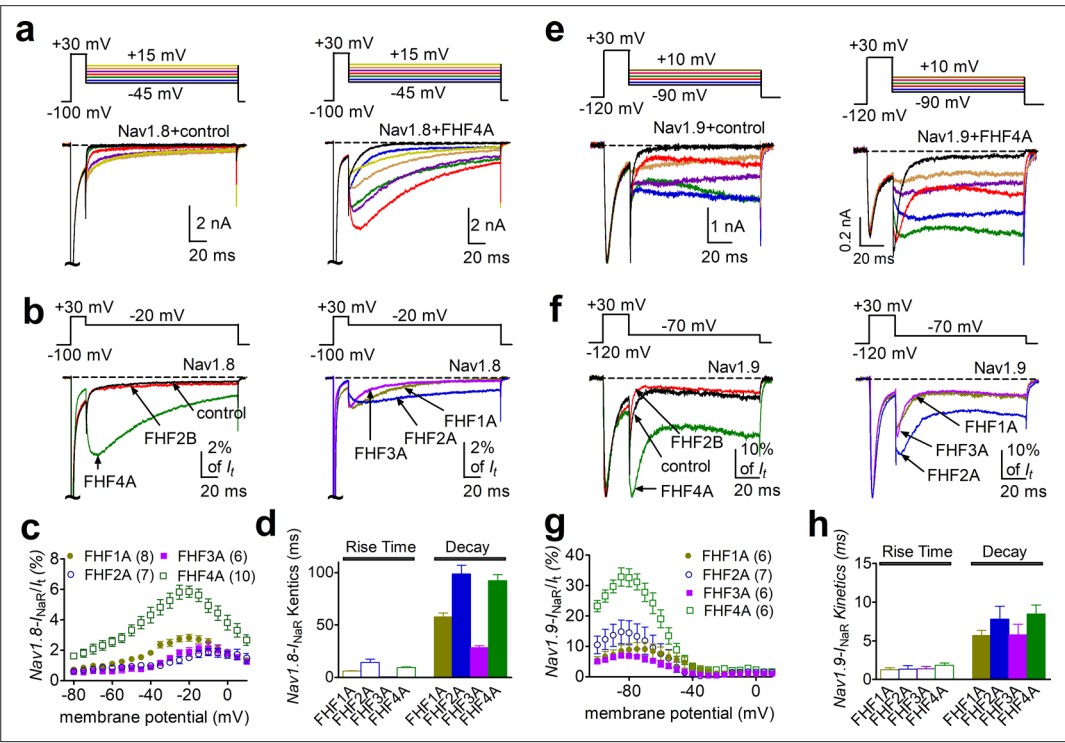

**Figure 2.** $I_{NaR}$ were produced by recombinant Nav1.8 and Nav1.9 coexpressed with FHF2A or FHF4A in heterologous systems. (**a, e**) Family of representative current traces recorded from cells expressing Nav1.8 or Nav1.9 that generated $I_{NaR}$ in the presence of FHF4A (right) and that did not in the absence of any fibroblast growth factor homologous factors (FHFs) (control, left). Currents were elicited by a standard resurgent current protocol shown in the inset. (**b, f**) Overlay of single-current traces of Nav1.8–Nav1.9 elicited by the protocol (inset) in the absence (control, black) or presence of FHF2B (red), FHF1A (yellow), FHF2A (blue), FHF3A (purple), and FHF4A (green). (**c, g**) Voltage dependence of the relative Nav1.8 and Nav1.9 $I_{NaR}$ mediated by FHF1A–FHF4A. Nav1.8 and Nav1.9 $I_{NaR}$ are normalized to the peak transient currents elicited at 0 mV and –30 mV, respectively. (**d, h**) The rise time (*time to peak*) and time constants of the decay kinetics of FHF-mediated $I_{NaR}$ in Nav1.8 and Nav1.9. While cells expressing Nav1.8 were held at –100 mV, cells expressing Nav1.9 were at –120 mV. The number of separate cells tested is indicated in parentheses. Data points are shown as mean ± SE.

The online version of this article includes the following figure supplement(s) for figure 2:

**Figure supplement 1.** Extreme slow nondecay currents were caused by slow recovery from inactivation of Nav1.9 'window currents.'.

We next examined the effects of FHFs on $I_{NaR}$ generation. FHF2A and FHF4A induced robust $I_{NaR}$ from Nav1.8 and Nav1.9 (*Figure 2a–h*). However, under control conditions and with coexpression of FHF2B, the repolarization pulses only elicited classic tail currents, which arise nearly instantaneously and decay rapidly, in Nav1.8 and Nav1.9 (*Figure 2a (left), b, e (left), and f*). This is the first demonstration of $I_{NaR}$ generation in a heterologous expression system without inclusion of an exogenous peptide in the intracellular solution. The FHF-mediated Nav1.8 $I_{NaR}$ peaked at –20 to –10 mV and could be observed at repolarization pulses ranging from +5 to –80 mV, while the FHF-mediated Nav1.9 $I_{NaR}$ displayed a more hyperpolarized voltage dependence, peaking at –85 mV and observed at repolarizing potentials ranging from –55 to –100 mV (*Figure 2e and g*). Moreover, the Nav1.8 $I_{NaR}$ induced by FHF4A were fourfold larger than those by FHF2A (*Figure 2C*, relative amplitudes of the peak transient current: FHF2A: 1.5% ± 0.2%; FHF4A: 5.9% ± 0.4%), and the Nav1.9 $I_{NaR}$ mediated by FHF4A was twofold larger than those induced by FHF2A (FHF2A: 14.8% ± 3.9%; FHF4A: 32.6% ± 3.0%). The kinetics of Nav1.8 $I_{NaR}$ mediated by A-type FHF are slow, with a slow onset and slow decay. The time to peak and the decay time constant for the FHF4A-mediated $I_{NaR}$ elicited at –20 mV were 9.63 ± 0.61 ms and 85.97 ± 5.29 ms, respectively (*Figure 2d*), similar to the TTX-resistant $I_{NaR}$ previously recorded from DRG neurons (*Dib-Hajj et al., 2015*). In contrast, FHF-mediated Nav1.9 $I_{NaR}$ exhibit fast onset and decay kinetics. At –70 mV, near the physiological resting membrane potential of DRG neurons, the time to peak and the decay time constant for FHF4A-mediated Nav1.9 $I_{NaR}$ were 1.92 ± 0.12 ms and 8.09 ± 1.31 ms, respectively (*Figure 2h*). It is noteworthy that Nav1.9 produced a nondecaying inward current following the $I_{NaR}$ (control, *Figure 2e–f*). This nondecaying current activated extremely slowly during 100 ms voltage pulses, occurred in the absence and presence of FHFs, and persisted even when the repolarization pulse was extended to 1000 ms (*Figure 2—figure supplement 1a*). Importantly, the current-voltage curve almost completely overlapped that of a predicted 'window current' formed by superimposition of steady-state activation and inactivation curves (*Figure 2—figure supplement 1b–d*). Since 'window current' typically results in persistent current (*Attwell et al., 1979*), we suggest that these nondecaying currents result from a slow recovery from inactivation of Nav1.9 currents. Regardless, these data indicate for the first time that Nav1.9 channels can generate a novel $I_{NaR}$ distinct from those generated by Nav1.8 and TTX-sensitive VGSCs.

In addition to FHF2A and FHF4A, FHF1A and FHF3A are also A-type FHFs and are predominantly expressed in the CNS (*Liu et al., 2001*; *Goetz et al., 2009*). Because ectopic Nav1.8 expression has been observed in CNS neurons in multiple sclerosis (*Black et al., 2000*), we examined whether FHF1A and FHF3A might induce $I_{NaR}$ in Nav1.8 and Nav1.9 as well. In addition to FHF2A and FHF4A, FHF1A and FHF3A both induced $I_{NaR}$ in Nav1.8 (FHF1A, 2.8% ± 0.2%; FHF3A, 1.9% ± 0.3%) and Nav1.9 (FHF1A, 9.0% ± 2.3%; FHF3A, 5.8% ± 1.1%). These $I_{NaR}$ displayed a voltage dependence of activation similar to those observed with FHF2A and FHF4A (*Figure 1b–f*). Based on the relative amplitudes of the generated $I_{NaR}$, the rank order of the ability of the four A-type FHFs to mediate Nav1.8 $I_{NaR}$ is FHF4A > FHF1A > FHF3A ≈ FHF2A, while the rank order for Nav1.9 $I_{NaR}$ generation is FHF4A > FHF2A > FHF1A > FHF3A.

## F2A/F4A peptides fully reconstituted $I_{naR}$

FHF2A, but not FHF2B, induces robust $I_{NaR}$ from Nav1.8 and Nav1.9 (*Figure 2b and f*). Intriguingly, FHF2A and FHF2B differ only in their N-terminus due to the alternative splicing of exon 1 (*Figure 3a*). Moreover, a peptide derived from amino acids 2–21 of the FHF2A N-terminus has been shown to induce long-term inactivation of Nav1.6 channels (*Dover et al., 2010*). We hypothesized that the same region of the N-terminal tail is the critical molecular component necessary for $I_{NaR}$ induction by A-type FHFs. To test this hypothesis, we intracellularly applied a 20-residue peptide (F2A or F4A), derived from the N-terminal residues 2–21 of FHF2A or FHF4A. We asked whether these peptides could reconstitute A-type FHF-mediated $I_{NaR}$ observed with coexpression of full-length FHF2A/FHF4A (*Figure 3a*). In the presence of 1 mM F2A or F4A, both Nav1.8 and Nav1.9 generated $I_{NaR}$. The relative amplitudes were 1.5% ± 0.1% and 2.5% ± 0.4% of peak transient current in Nav1.8 at –15 mV (*Figure 3b and c*), and 18.0% ± 3.1% and 10.3% ± 1.4% in Nav1.9 at –85 mV, respectively (*Figure 3f and g*). The $I_{NaR}$ retained the kinetics and voltage dependence of activation as observed with full-length FHF2A and FHF4A (*Figure 2c and g*). On the other hand, both F2A and F4A significantly decreased the inactivation time constant of the transient currents of Nav1.8 and Nav1.9 evoked by a 20 ms pre-pulse to +30 mV (*Figure 3d and g*), suggesting that both F2A and F4A serve as open

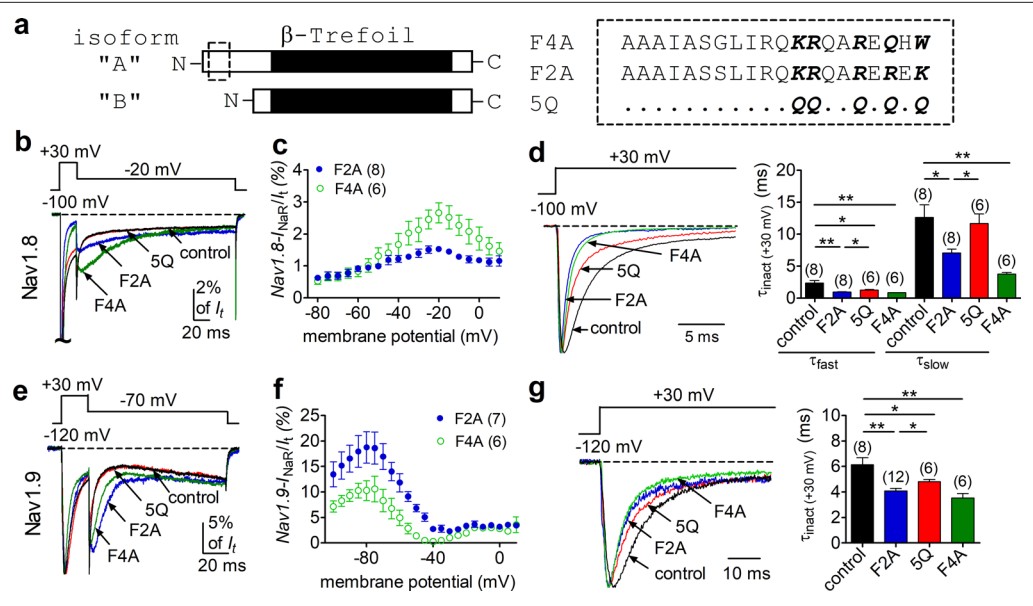

**Figure 3.** The peptides F2A and F4A fully reconstituted FHF2A/FHF4A-induced $I_{NaR}$ in Nav1.8 and Nav1.9 in heterologous systems. (**a**) Schematic diagram of A- and B-type fibroblast growth factor homologous factors (FHFs) (left). The amino acid sequences of short peptides located at N terminus of FHF2A and FHF4A are shown (right). Five positively charged residues of interest are highlighted in bold. 5Q is a mutant of F2A, in which five positive residues are replaced by Gln (**Q**). The residues conserved in F2A are indicated as dots. (**b**) Overlay of representative Nav1.8 $I_{NaR}$ traces in the absence (control, black) and presence of F2A (blue), 5Q (red), or F4A (green). (**c**) Voltage dependence of the relative F2A- and F4A-induced Nav1.8 $I_{NaR}$. Nav1.8 $I_{NaR}$ are normalized to the peak transient current elicited at 0 mV. (**d**) Decay time constants ($\tau$, right) of transient Nav1.8 currents (left) at +30 mV. The time constants ($\tau_{fast}$, $\tau_{slow}$) were well fitted by a double exponential function. $\tau_{fast}$: control, 2.35 ± 0.40 ms; F2A, 0.92 ± 0.09 ms (p=0.0037 vs. control); 5Q, 1.26 ± 0.07 ms (p=0.0339 vs. F2A); F4A, 0.83 ± 0.04 ms (p=0.0071 vs. control). $\tau_{slow}$: control, 12.65 ± 1.95 ms; F2A, 7.07 ± 0.60 ms (p=0.0161 vs. control); 5Q, 11.67 ± 1.52 ms (p=0.0065 vs. F2A); F4A, 3.72 ± 0.28 ms (p=0.0021 vs. control). (**e**) Overlay of Nav1.9 $I_{NaR}$ traces in the absence (control, black) and presence of F2A (blue), 5Q (red), or F4A (green). (**f**) Voltage dependence of the relative F2A- and F4A-induced Nav1.9 $I_{NaR}$. (**g**) Decay time constants ($\tau$, right) of transient Nav1.9 currents (left) at +30 mV. The time constants were fitted well by a single-exponential function. Cells expressing Nav1.8 or Nav1.9 were held at –100 mV or –120 mV, respectively. All $I_{NaR}$ of Nav1.8 or Nav1.9 were normalized to the peak transient current at –40 mV or at 0 mV, respectively. The concentrations of F2A, 5Q, and F4A all are 1 mM. Filled and open circles represent FHF2A and FHF4A, respectively. The number of separate cells tested is indicated in parentheses. *p<0.05; **p<0.01.

channel blockers of Nav1.8 and Nav1.9. This is consistent with the previous reports that F2A induces open-channel block in Nav1.5 and Nav1.6 (*Venkatesan et al., 2014*; *Dover et al., 2010*).

To further investigate the roles of these peptides in $I_{NaR}$ generation, we employed an F2A mutant (*Dover et al., 2010*) in which five positively charged residues (K1/R2/R3/R4/K5) are substituted with the neutral residue glutamine (5Q, *Figure 3a*). In *Figure 3b and e*, the mutant 5Q peptide failed to induce $I_{NaR}$ in Nav1.8 or Nav1.9: the currents elicited during the repolarization pulse almost overlapped in the absence (control) and presence of 1 mM 5Q. Consistent with this finding, the transient current inactivated more slowly at +30 mV with 5Q than with F2A (*Figure 3d and g*). Interestingly, the transient current still inactivated faster than under control conditions, suggesting that 5Q may still bind to VGSCs, but with lower affinity compared to F2A. These results suggest that the five positively charged residues in A-type FHFs are critical components for inducing $I_{NaR}$.

## FHF4A-mediated Nav1.8 $I_{NaR}$ in sensory neurons

DRG neurons show expression of FHF4A along with Nav1.8 and Nav1.9. Our results in heterologous systems showed that FHF4A was most capable among the A-type FHF isoforms at inducing $I_{NaR}$ with Nav1.8 and Nav1.9. Therefore, we next asked if FHF4A mediates $I_{NaR}$ in primary neurons. We are able to isolate Nav1.9 $I_{NaR}$ in DRG neurons (*Figure 4*). However, while these unique $I_{NaR}$ are strikingly similar

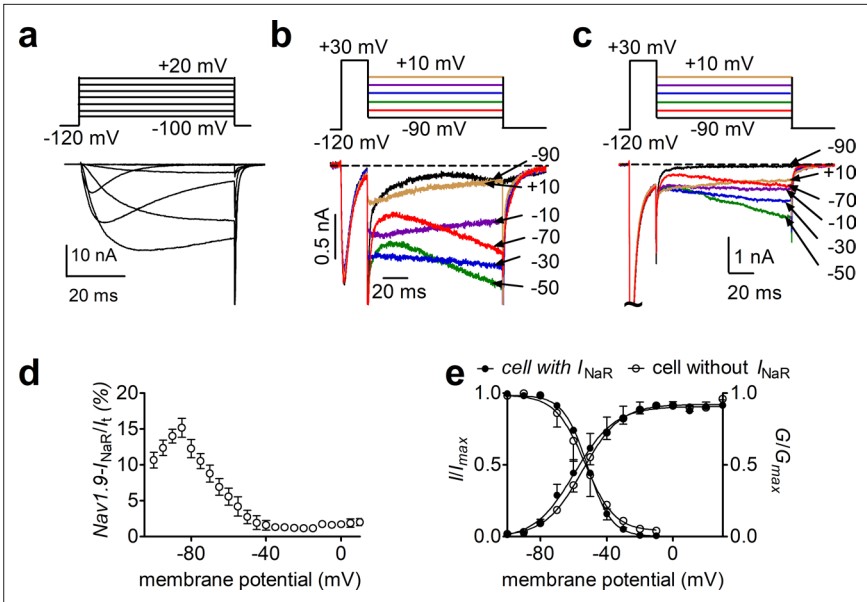

**Figure 4.** Nav1.9 $I_{NaR}$ generated from Nav1.8 knockdown dorsal root ganglion (DRG) neurons. (**a**), Typical Nav1.9 current traces induced by the protocol (inset), in which cells were subjected to 50 ms depolarization of potentials ranging from −120 to +40 mV with a 10 mV increment from a holding potential of −120 mV. (**b, c**) Representative current traces recorded from DRG neurons that did (**b**) and that did not (**c**) generate $I_{NaR}$. Currents were elicited by a standard $I_{NaR}$ protocol (inset), where cells were initially depolarized to +30 mV for 20 ms, then followed by a 100 ms hyperpolarizing potential ranging from +10 to −100 mV. (**d**), Voltage dependence of Nav1.9 $I_{NaR}$ shown in (**b**). All $I_{NaR}$ were normalized to peak transient current. (**e**), Steady-state activation and inactivation measured on DRG neurons with or without $I_{NaR}$.

to those recorded when A-type FHFs are coexpressed with Nav1.9 in HEK293 cells, the endogenous Nav1.9 $I_{NaR}$ are only evident in a small subset of DRG neurons. On the other hand, in our previous work Nav1.8 type $I_{NaR}$ could be recorded from the majority of DRG neurons expressing endogenous Nav1.8 currents and almost all DRG neurons expressing recombinant Nav1.8 (*Xiao et al., 2019*). We therefore next focused on interrogating the role of Nav1.8 $I_{NaR}$ in DRG neurons.

Consistent with our previous observation (*Xiao et al., 2019*), 13/14 small-diameter DRG neurons transfected with a scrambled FHF4A shRNA were found to generate Nav1.8 $I_{NaR}$. The largest $I_{NaR}$ was attained at −15 mV, with an average relative amplitude of 2.1% ± 0.3% of the peak transient TTX-resistant sodium current. The time to peak and the decay time constant for the current elicited at −15 mV were 45.0 ± 4.4 ms and 546.3 ± 43.2 ms, respectively. These results were identical to those seen in DRG neurons, without scrambled shRNA, in our previous work (*Xiao et al., 2019*), suggesting that the scrambled shRNA did not alter Nav1.8 $I_{NaR}$. The efficiency of FHF4shRNA-mediated knockdown was determined using a monoclonal antibody specific to FHF4, which has been validated in heterologous systems in our laboratory. In *Figure 5a and b*, FHF4shRNA reduced FHF4 expression by 73.1% (p<0.0001). FHF4 knockdown did not significantly alter current density, voltage dependence of activation or recovery rate from inactivation of Nav1.8 currents in DRG neurons, but caused a hyperpolarizing shift of 12 mV in the voltage dependence of steady-state inactivation (p<0.0001; *Figure 5c–e*, *Table 2*). FHF4 knockdown considerably decreased the proportion of DRG neurons producing $I_{NaR}$ (9/18 cells vs. 13/14 scramble cells; p=0.0095, $\chi^2$ test; *Figure 5g*). Furthermore, in those DRG neurons with $I_{NaR}$, FHF4 knockdown did not modify the voltage dependence of activation of Nav1.8 $I_{NaR}$, but reduced the relative amplitude by about 42% (FHF4shRNA, 1.2% ± 0.2%; p<0.05; *Figure 5h*). Although our previous work showed that Navβ4 can contribute to generation of Nav1.8 $I_{NaR}$ in DRG neurons (*Xiao et al., 2019*), the reduction here was Navβ4 independent because FHF4 knockdown did not significantly change Navβ4 expression level in our immunostaining experiments (*Figure 5—figure supplement 1*). Therefore, our data indicate that FHF4A is a major producer of Nav1.8 $I_{NaR}$ in small-diameter DRG neurons. The remaining $I_{NaR}$ after FHF4 knockdown are possibly mediated by residual FHF4A, endogenous FHF2A, or endogenous Navβ4.

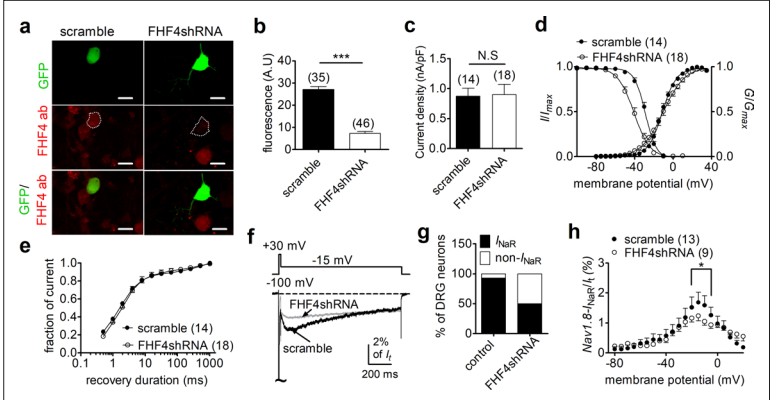

**Figure 5.** FHF4 knockdown reduced the ability of Nav1.8 to generate $I_{NaR}$ in rat dorsal root ganglion (DRG) neurons. (**a**) Immunofluorescent reactions showed expression levels of FHF4 in DRG neurons. Dashed lines show the shape of transfected DRG neurons. Scale bars, 50 μm; ab, antibody. (**b**) Summary of fluorescence in DRG neurons transfected with the scrambled shRNA or FHF4shRNA (p<0.0001). (**c**) FHF4 knockdown did not significantly alter Nav1.8 current density (p=0.9116). (**d**) FHF4 knockdown shifted voltage dependence of steady-state inactivation to more negative potentials (p<0.0001), but did not affect activation (p=0.9116). (**e**) FHF4 knockdown did not significantly impair the recovery rate from inactivation The time constants estimated from single-exponential fits were 2.92 ± 0.53 ms (scramble) and 4.00 ± 1.01 ms (FHF4shRNA, p=0.3905),. (**f**) $I_{NaR}$ traces recorded from small-diameter DRG neurons transfected with scramble or FHF4shRNA. (**g**) FHF4 knockdown decreased the percentage of DRG neurons to generate Nav1.8 $I_{NaR}$ (p<0.0001). (**h**) Voltage dependence of the relative Nav1.8 $I_{NaR}$ in DRG neurons treated with scramble and FHF4shRNA. Filled and open circles represent scramble and FHF4shRNA, respectively. The number of separate cells tested is indicated in parentheses. N.S., not significant; *p<0.05; ***p<0.0001.

The online version of this article includes the following figure supplement(s) for figure 5:

**Figure supplement 1.** FHF4 knockdown did not influence Navβ4 expression in dorsal root ganglion (DRG) neurons.

## FHF4A-mediated Nav1.8 $I_{NaR}$ regulated sensory neuron excitability

We next explored the impact of FHF4A-mediated $I_{NaR}$ on neuronal excitability. Previous studies have shown that FHFs profoundly modulate the activities of TTX-sensitive VGSCs in DRG neuron (*Barbosa et al., 2017*; *Venkatesan et al., 2014*; *Dover et al., 2010*); therefore, we measured excitability of small-diameter DRG neurons in the presence of 500 nM TTX, which blocks all TTX-sensitive VGSCs and removes this confounding variable. FHF4 knockdown did not change resting membrane potential, input resistance, or rheobase of action potential firing (scramble, –54.9 ± 1.1 mV, 463.3 ± 38.5 MΩ, 1.26 ± 0.21 nA; FHF4shRNA, –57.2 ± 1.6 mV, 548.7 ± 66.9 MΩ, 1.22 ± 0.14 nA; p>0.05; *Figure 6a–c*), but narrowed single-evoked action potentials. The average action potential durations measured under scramble and FHF4shRNA were 17.94 ± 2.63 ms and 10.54 ± 1.19 ms (p=0.0153; *Figure 6d*), respectively. With 2 s injected currents greater than 300 pA, the FHF4-knockdown DRG neurons displayed significantly fewer action potentials than the scramble-treated neurons (*Figure 6e and f*).

We then tested whether F4A peptide could reverse the FHF4A knockdown-mediated effects on $I_{NaR}$ and neuronal excitability. Intracellular application of 1 mM F4A did not significantly alter current density, voltage dependence of activation, steady-state inactivation, or recovery rate from inactivation of Nav1.8 currents in FHF4shRNA-treated DRG neurons (*Figure 6g–i*, *Table 2*). Although F4A peptide

**Table 2.** Gating properties of Nav1.8 in dorsal root ganglion (DRG) neurons.
Midpoint voltages of the steady-state activation and inactivation curves in *Figures 5 and 6* were determined with a standard Boltzmann distribution fit. @p<0.001 vs. respective control condition. The number of separate cells tested is indicated in parentheses.

| $V_{1/2}$ **(mV)** | Control | FHF4shRNA | +F4**A** |
|---|---|---|---|
| Activation | –11.1 ± 1.1 (14) | –10.7 ± 2.9 (18) | –9.3 ± 3.5 (10) |
| Inactivation | –28.5 ± 1.0 (14) | –40.4 ± 2.0@ (18) | –35.9 ± 1.2@ (10) |

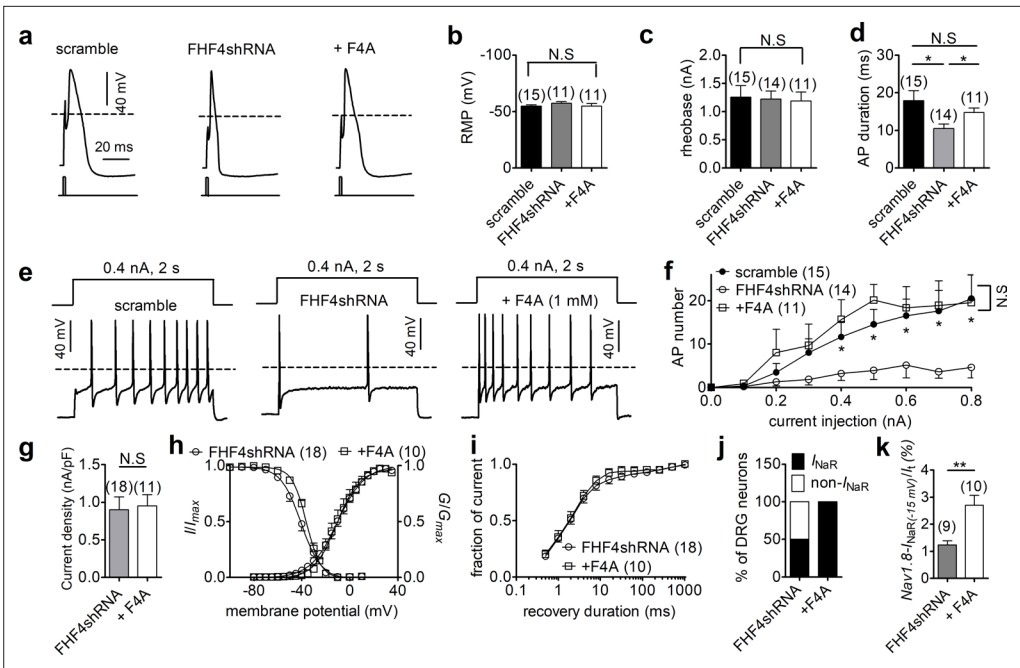

**Figure 6.** FHF4shRNA-mediated reduction in dorsal root ganglion (DRG) neuron excitability was rescued by the F4A peptide. (**a**) Typical single-action potentials elicited by a 1 ms current injection. (**b**) Resting membrane potentials under scramble, FHF4shRNA, and FHF4shRNA + F4A (p=0.6149, one-way ANOVA). (**c**) Summary of rheobase (p=0.9673, one-way ANOVA). (**d**) Summary of action potential duration (APD90). The durations were 17.94 ± 2.63 ms (scramble), 10.54 ± 1.19 ms (FHF4shRNA, p=0.0153 vs. control), and 14.76 ± 1.22 ms (+F4A, p=0.0233 vs. FHF4shRNA and p=0.3011 vs. control), respectively. (**e**) Typical action potential trains elicited by a 2 s injection of 400 pA current. (**f**) Summary of the number of action potentials elicited by a 2 s injection of currents ranging from 0 to 800 pA. (**g**) F4A did not alter Nav1.8 current density (p=0.8428). (**h**) Voltage dependence of activation and steady-state inactivation of Nav1.8 before and after addition of F4A in FHF4shRNA-treated DRG neurons (activation: p=0.8160; inactivation: p=0.0332). (**i**) F4A did not impair the recovery rate from Nav1.8 inactivation in FHF4shRNA-treated DRG neurons. The time constants estimated from single exponential fits were 4.00 ± 1.01 ms (FHF4shRNA) and 2.92 ± 0.42 ms (+F4A, p=0.5826), respectively. (**j**) F4A increased the percentage of FHF4shRNA-treated DRG neurons to generate Nav1.8 $I_{NaR}$ (p=0.0066). (**k**) F4A increased the relative amplitude of Nav1.8 $I_{NaR}$ in FHF4shRNA-treated DRG neurons (p=0.0027). In (**a–k**), the concentration of F4A is 1 mM. Filled circles, open circles, and open squares represent scramble, FHF4shRNA, and F4A, respectively. The number of separate cells tested is indicated in parentheses. The $V_{1/2}$ values measured in (**h**) are summarized in **Table 2**. N.S., not significant; *p<0.05; ***p<0.001.

did not reverse the negative shift in the voltage dependence of steady-state inactivation caused by FHF4 knockdown (shown in **Figure 5d**), F4A peptide did rescue the FHF4-knockdown-mediated decrease in $I_{NaR}$: 10/10 DRG neurons tested yielded Nav1.8 $I_{NaR}$ (p=0.0066; $\chi^2$ test; **Figure 6j**). The average relative amplitude measured at –15 mV increased from 1.2% ± 0.2% (FHF4shRNA) to 2.7% ± 0.4% (F4A; p=0.0027; **Figure 6k**), similar to the amplitude yielded under the scramble condition. F4A peptide did not change the resting membrane potential, input resistance, or rheobase in FHF4shRNA-treated DRG neurons (+F4A, –54.8 ± 2.3 mV, 502.8 ± 92.9 MΩ, 1.19 ± 0.15 nA; p>0.05 vs. scramble and FHF4shRNA; **Figure 6b and c**), but significantly broadened action potentials (average duration of 14.76 ± 1.22 ms; p=0.0233; **Figure 6d**). F4A peptide increased the number of action potentials elicited by 2 s injected currents of 400 pA (**Figure 6e**). Finally, the FHF4shRNA-transfected DRG neurons treated with F4A peptide could fire action potentials at almost the same frequency as the scramble-transfected neurons (**Figure 6f**), demonstrating that the loss of sensory neuron excitability by FHF4 knockdown can be rescued by F4A. Therefore, our data clearly illustrate that A-type FHF is a critical molecule in small-diameter DRG neurons and that A-type FHF determines neuronal excitability via $I_{NaR}$ generation.

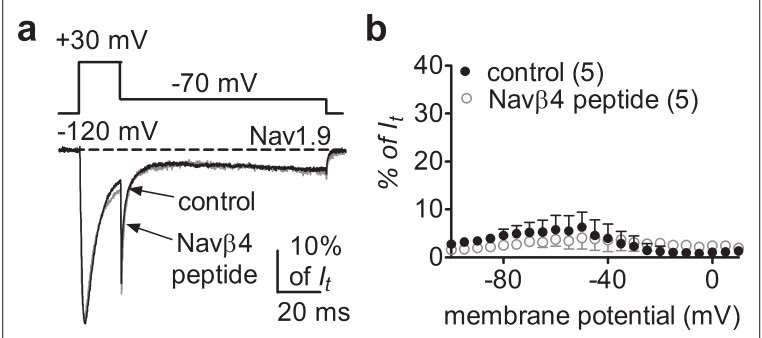

**Figure 7.** Navβ4 peptide did not induce Nav1.9 $I_{NaR}$ in HEK293 cells. (**a**) Overlay of normalized current traces elicited by a resurgent protocol (inset) in the absence (control, gray) and presence of 200 μM Navβ4 peptide (black). (**b**) Voltage dependence of the relative currents. Filled and open circles represent control and Navβ4 peptide, respectively.

## Navβ4 does not elicit Nav1.9 $I_{naR}$

The Nav1.9 $I_{NaR}$ identified in HEK293 cells cotransfected with Nav1.9 and A-type FHFs is distinct from the $I_{NaR}$ observed with other VGSCs. As Navβ4 has been shown to induce $I_{NaR}$ in all of the other VGSC isoforms (Nav1.1–Nav1.8), we asked if Navβ4 also induces $I_{NaR}$ with Nav1.9. Multiple studies have failed to reconstitute $I_{NaR}$ in heterologous systems by coexpressing full-length Navβ4 with VGSC α-subunits (*Catterall et al., 2005*, *Lin et al., 2016*, *Liu et al., 2001*). However, a short peptide (KKLIT-FILKKTREK) derived from the Navβ4 C-terminus can induce $I_{NaR}$ generation in heterologous expression systems and also in primary neurons (*Grieco et al., 2005*; *Barbosa et al., 2015*). Here, we intracellularly applied Navβ4 peptide (200 μM) to investigate if Nav1.9 expressed in HEK293 cells could utilize the C-terminus of Navβ4 to generate $I_{NaR}$. Surprisingly, Navβ4 peptide did not induce Nav1.9 $I_{NaR}$. Only classic tail currents were observed with the Navβ4 peptide (p>0.05; *Figure 7a and b*), indicating that Navβ4 is not capable of mediating $I_{NaR}$ in Nav1.9.

## An inner pore residue impairs Navβ4, but not A-type FHFs, $I_{naR}$

Nav1.9 exhibits low (42–53%) sequence similarity to other mammalian VGSC subtypes. We hypothesized that nonconserved pore residues, especially positive residues, might prevent the positively charged Navβ4 peptide from binding to the Nav1.9 inner pore. Sequence analysis identified K799, residing in the II-S6 segment of Nav1.9, as a promising candidate for such prevention (*Figure 8a*). The corresponding residue in all other VGSC isoforms is an asparagine. As described previously (*Lin et al., 2016*), the K799N mutation does not significantly alter Nav1.9 gating properties (*Figure 8b*, *Table 3*). Interestingly, the K799N mutation greatly enhanced the ability of Navβ4 peptide (200 μM) to mediate Nav1.9 $I_{NaR}$ in response to a depolarizing voltage of +100 mV. *Figure 8c* shows that Navβ4 peptide mediated $I_{NaR}$ in K799N channels with a fast onset/decay kinetics and a hyperpolarized voltage dependence of activation similar to A-type FHF-mediated Nav1.9 $I_{NaR}$. The relative amplitude is 6.6% ± 0.5% at –85 mV (*Figure 8d*). Intriguingly, the K799N mutation did not alter F2A-mediated $I_{NaR}$ (Nav1.9, 17.0% ± 2.5%; K799N, 18.1% ± 3.5%; p>0.05; *Figure 8e and f*).

To further confirm the role of this residue in modulating VGSCs $I_{NaR}$, we constructed reverse mutations at corresponding positions in Nav1.5 and Nav1.7 (*Figure 8a*). The reverse mutation N927K did not influence gating properties of Nav1.5 (*Figure 8b*, *Table 3*), but reduced Nav1.5 $I_{NaR}$ induced by the presence of 200 μM Navβ4 peptide by 92% (Nav1.5, 17.2% ± 2.1%; N927K, 1.3% ± 0.1%; p<0.0001; *Figure 8g and h*). A substantial reduction (~85%) was also observed for the N945K mutation in Nav1.7 (*Figure 8—figure supplement 1*). In contrast, the N927K mutation did not impair the ability of Nav1.5 to generate $I_{NaR}$ mediated by full-length FHF2A (Nav1.5, 0.24% ± 0.03%; N927K, 0.23% ± 0.03%; p>0.05; *Figure 8i and j*). Collectively, these results indicate that the residue at position 799 in Nav1.9 is involved in VGSC interaction with Navβ4. Although K799 in Nav1.9 is a major determinant of Nav1.9 resistance to the Navβ4 peptide, it may not be the only factor involved in Nav1.9 resistance. Furthermore, because changes at this position did not alter A-type FHF mediated $I_{NaR}$, we propose that A-type FHFs and Navβ4 do not share identical binding determinants in the pore of VGSCs.

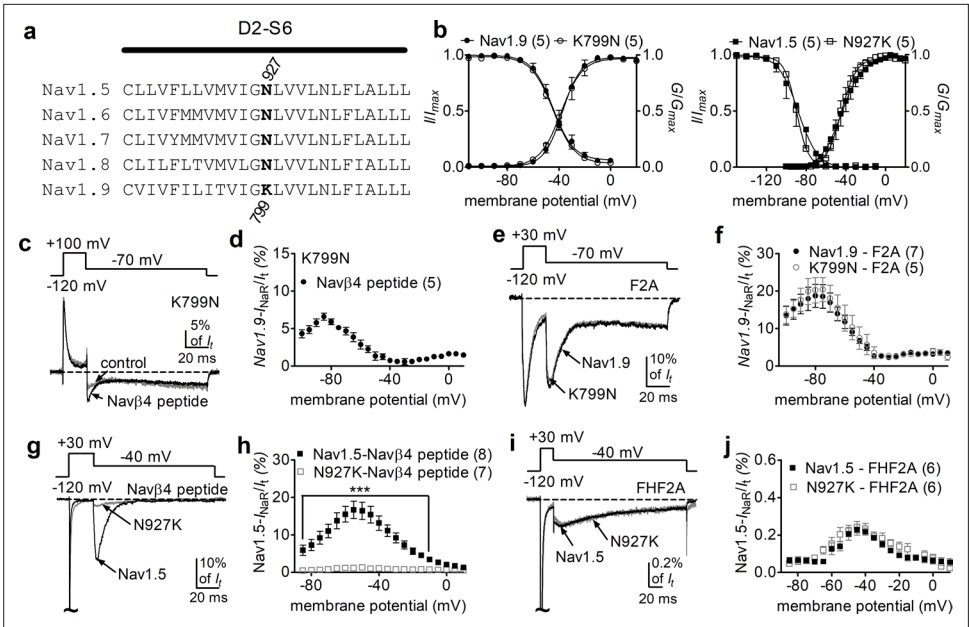

**Figure 8.** The residue at position 799 in Nav1.9 was crucial for voltage-gated sodium channel (VGSC) sensitivity to Navβ4. (**a**) Sequence alignment of domain II S6 segments of Nav1.5–Nav1.9. The position of the residues of interest is indicated in bold and designated with a number. (**b**) The K799N mutation and the reversal mutation N927K did not significantly alter steady-state activation or inactivation of Nav1.9 (circles, right) and Nav1.5 (squares, left), respectively. (**c**) The Nav1.9 mutant K799N generated $I_{NaR}$ in the presence of 200 μM Navβ4 peptide (black). Control, gray. (**d**) Voltage dependence of the relative $I_{NaR}$ in the Nav1.9 mutant K799N (filled circles). (**e**) Typical $I_{NaR}$ traces recorded from Nav1.9 (black) and the mutant K799N (gray) in the presence of 1 mM F2A. (**f**) Comparison of the relative F2A-induced $I_{NaR}$. Filled and open circles represent Nav1.9 and the mutant K799N, respectively. (**g**) Typical $I_{NaR}$ traces recorded from Nav1.5 (black) and the mutant N927K (gray) in the presence of 200 μM Navβ4 peptide. (**h**) Voltage dependence of the relative $I_{NaR}$ in Nav1.5 (filled squares) and the mutant N927K (open squares). (**i**) Typical $I_{NaR}$ traces recorded from Nav1.5 (black) and the mutant N927K (gray) in the presence of FHF2A. (**j**) Comparison of the relative FHF2A-induced $I_{NaR}$ in Nav1.5 (filled squares) and the mutant N927K (open squares). In (**c, e, g, i**), $I_{NaR}$ were elicited by the protocols shown in the inset. In (**b, c, d, g, h**), 500 μM GTP-γ-S was added for Nav1.9 and K799N cells in the pipette solution. F2A and Navβ4 peptide were applied in peptide solution. The number of separate cells tested is indicated in parentheses. ***p<0.005.

The online version of this article includes the following figure supplement(s) for figure 8:

**Figure supplement 1.** The N945K mutation substantially reduced Navβ4-mediated Nav1.7 $I_{NaR}$ in HEK293 cells.

## FHF2A-mediated Nav1.5 and Nav1.7, but not Nav1.6, $I_{NaR}$ in heterologous system

Finally, we asked if other VGSC isoforms share the FHF mechanism of $I_{NaR}$ generation. We studied Nav1.5, Nav1.6, and Nav1.7 because they are coexpressed with FHF2A or FHF4A in cardiac myocytes and neurons (*Li et al., 2002*; *Wang et al., 2011a*; *Yan et al., 2014*; *Barbosa et al., 2017*; *White et al., 2019*). Coexpression of FHF2A with Nav1.5 or Nav1.7 induced $I_{NaR}$ (*Figure 9a–f*), with a voltage dependence of activation more negative than Nav1.8 but more positive than Nav1.9 $I_{NaR}$ (*Figure 1c*

**Table 3.** Gating properties of wild-type Nav1.5, the mutant N927K, wild-type Nav1.9, and the mutant K799N.

Midpoint voltages of the steady-state activation and inactivation curves in *Figure 8* were determined with a standard Boltzmann distribution fit. All changes are not statistically significant vs. respective wild-type condition. The number of separate cells tested is indicated in parentheses.

| $V_{1/2}$ (mV) | Nav1.5wt | N927K | Nav1.9wt | K799N |
|---|---|---|---|---|
| Activation | −44.6 ± 4.4 (5) | −44.0 ± 2.9 (5) | −37.1 ± 3.3 (5) | −37.5 ± 4.1 (5) |
| Inactivation | −88.5 ± 6.4 (5) | −89.7 ± 0.9 (5) | −45.7 ± 1.9 (7) | −44.0 ± 1.5 (5) |

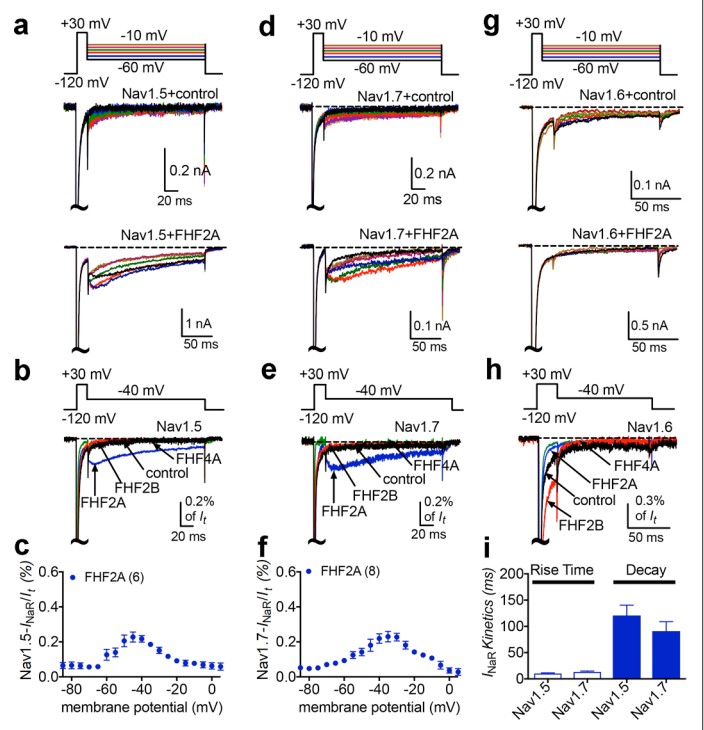

**Figure 9.** $I_{NaR}$ were produced by recombinant Nav1.5 and Nav1.7 coexpressed with FHF2A in heterologous systems. (**a, d, e**) Family of representative current traces recorded from cells expressing Nav1.5, Nav1.7, or Nav1.6 in the presence of FHF2A (below) and that did not in the absence of any fibroblast growth factor homologous factors (FHFs) (control, upper). Currents were elicited by a standard $I_{NaR}$ protocol shown in the inset. (**b, e, h**) Overlay of single-current traces of Nav1.5–Nav1.7 elicited by the protocol (inset) in the absence (control, black) or presence of FHF2B (red), FHF2A (blue), and FHF4A (green). (**c, f**) Voltage dependence of the relative Nav1.5 and Nav1.7 $I_{NaR}$ mediated by FHF2A. (**i**) The rise time and time constants of the decay kinetics of FHF2A-mediated $I_{NaR}$ in Nav1.5 and Nav1.7. In (**c, f**), all $I_{NaR}$ were normalized to the peak transient current. In (**i**), time constants were obtained by fitting a single exponential function. Cells were held at −120 mV. The number of separate cells tested is indicated in parentheses. Data points are shown as mean ± SE.

*and f*). Maximal $I_{NaR}$ were attained at near −40 mV. The average relative amplitudes were at least sixfold smaller (Nav1.5, 0.22% ± 0.02%; Nav1.7, 0.27% ± 0.02%) than Nav1.8 and Nav1.9 $I_{NaR}$. The time to peak and the decay time constant for the FHF2A-mediated Nav1.5 $I_{NaR}$ were 10.2 ± 1.3 ms and 120.8 ± 19.6 ms at −40 mV, respectively (*Figure 9i*). FHF2A-mediated Nav1.7 $I_{NaR}$ displayed a similar rise and decay kinetics to Nav1.5 $I_{NaR}$ (*Figure 9i*). However, neither FHF2B nor FHF4A generated $I_{NaR}$ in Nav1.5 and Nav1.7 (*Figure 9b and e*). Neither FHF2A nor FHF4A induced generation of $I_{NaR}$ with Nav1.6 (*Figure 8g and h*). However, coexpression of FHF4A with Nav1.6 elicited long-term inactivation of Nav1.6 in ND7/23 cells (*Figure 10*), similar to that previously shown for coexpression of FHF2A with Nav1.6 in HEK293 cells (*Rush et al., 2006*). Furthermore, intracellular application of the F4A peptide did not induce $I_{NaR}$, only long-term inactivation similar to that induced by full-length FHF4A (*Figure 10*).

## Discussion

Resurgent sodium currents are critical regulators of central and peripheral neuron excitability. In this study, we identify A-type FHFs as direct mediators of TTX-resistant VGSC $I_{NaR}$. We show, for the first time, that coexpression of only two proteins, a full-length A-type FHF and a VGSC α-subunit, is sufficient in heterologous systems to reconstitute $I_{NaR}$. We show that short peptides derived from A-type FHF N-termini, the precise residues that induce long-term inactivation in other VGSCs (*Venkatesan et al., 2014*) can fully replicate the Nav1.8 and Nav1.9 TTX-resistant $I_{NaR}$. Importantly, we implicate A-type FHFs as major drivers of TTX-resistant $I_{NaR}$ in nociceptive DRG neurons.

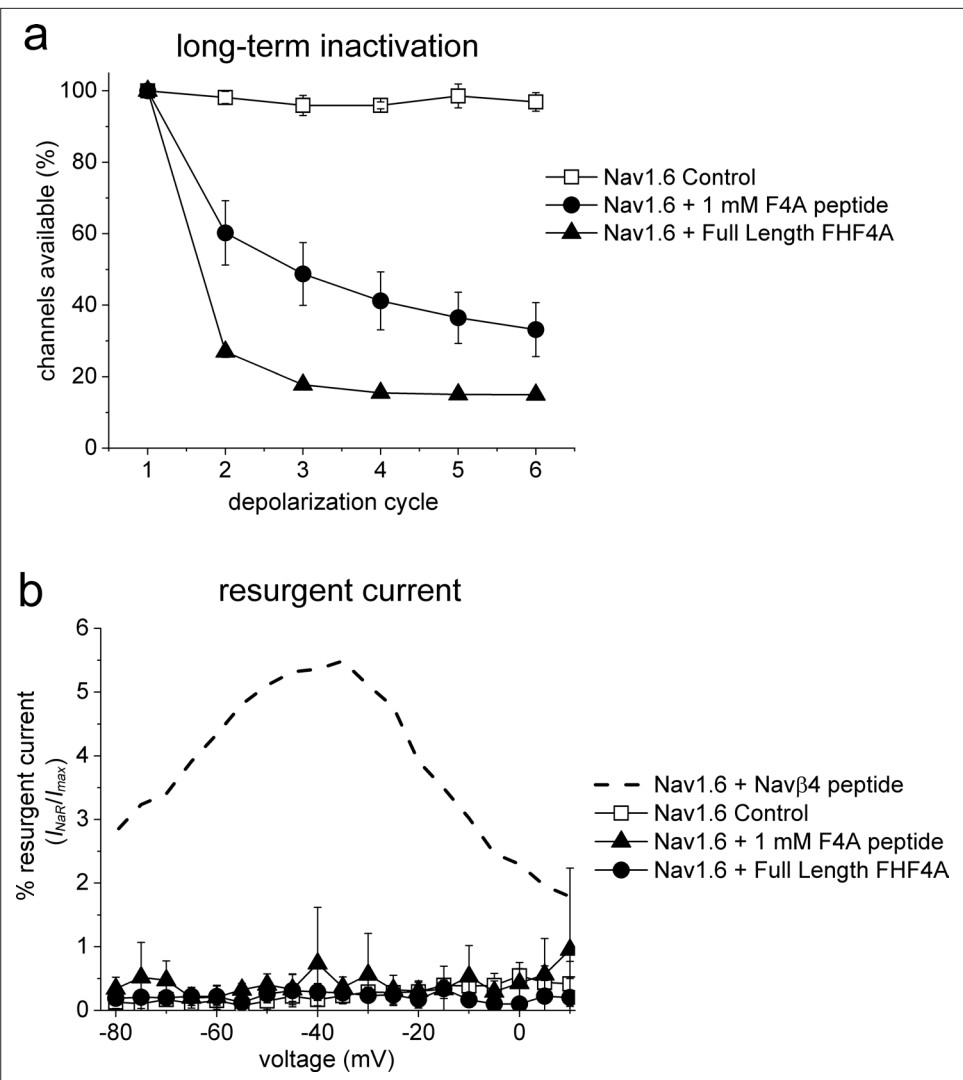

**Figure 10.** FHF4A induces long-term inactivation, not $I_{NaR}$, in Nav1.6 channels. HEK293 cells stably expressing human Nav1.6 were recorded under control conditions, after FHF4A transfection and with F4A peptide (1 mM) in the pipette solution. (**a**) Both the full-length FHF4A and the F4A peptide induced a substantial increase in long-term inactivation in response to a train of six –20 mV depolarizations at ~50 Hz. (**b**) Neither full-length FHF4A nor F4A peptide induced detectable $I_{NaR}$ in HEK293 cells expressing Nav1.6 channels. For comparison, data for Nav1.6 $I_{NaR}$ with Navβ4 peptide (200 mM) is shown with the dashed curve, adapted from ***Pan and Cummins, 2020***.

We also identified a novel TTX-resistant Nav1.9 $I_{NaR}$, which shows unique biophysical properties. The voltage dependence of activation of Nav1.9 $I_{NaR}$ is >40 mV more negative than previously described $I_{NaR}$ and these Nav1.9 currents exhibit faster onset/decay kinetics than Nav1.8 $I_{NaR}$. The ratio of $I_{NaR}$ to peak transient current is at least fivefold larger in Nav1.9 than in other VGSC isoforms. This is likely due to the extremely slow fast inactivation of Nav1.9 because destabilizing VGSC fast inactivation by disease mutations or toxins enhances $I_{NaR}$ generation (***Grieco and Raman, 2004***; ***Jarecki et al., 2010***; ***Bant et al., 2013***; ***Xiao et al., 2019***). Interestingly, while coexpression of A-types FHFs resulted in consistent Nav1.9 $I_{NaR}$ in HEK293 cells, only a small fraction of rodent DRG neurons exhibited Nav1.9 $I_{NaR}$. This might due to differences between rodent and human Nav1.9, or as a result of the complex modulation of Nav1.9 $I_{NaR}$ in sensory neurons.

Our data reveal a novel mechanism of $I_{NaR}$ generation independent of Navβ4. This is surprising because a substantial number of studies have implicated Navβ4 as a major determinant of $I_{NaR}$ in multiple VGSC subtypes (***Grieco et al., 2005***; ***Jarecki et al., 2010***; ***Bant and Raman, 2010***; ***Lewis and Raman, 2011***; ***Miyazaki et al., 2014***; ***Barbosa et al., 2015***; ***Patel et al., 2016***). Our mutagenesis

experiments show that an inner pore residue (K799) that is unique to Nav1.9 determines the inability of Navβ4 peptide to induce Nav1.9 $I_{NaR}$. The K799 residue is replaced by asparagine at the corresponding position (D2-S6) in Nav1.1–Nav1.8. The homology model of Nav1.9 shows that the side chain of K799 projects into the channel pore (*Lin et al., 2016*), and thus the positively charged residue may prevent the positively charged Navβ4 peptide from accessing its binding site in the Nav1.9 inner pore by electrostatic repulsion, which is consistent with previous findings that positive residues are crucial for Navβ4 peptide mediating $I_{NaR}$ (*Lewis and Raman, 2011*).

The most salient finding from this study is identification of A-type FHFs as novel $I_{NaR}$ mediators of Nav1.8 and Nav1.9 and, to a lesser extent, Nav1.5 and Nav1.7. A-type FHFs consist of a long N-terminus, an FGF-like β-trefoil core, and a short C-terminus (*Goldfarb, 2005*; *Figure 6a*). Here, we show that the long N-terminus of A-type FHFs, specifically residues 2–21, is the molecular component responsible for inducing TTX-resistant $I_{NaR}$. While the F2A and F4A peptides show limited sequence similarity to the Navβ4 peptide, they all exhibit similar patterns of interactions with VGSCs. The F2A, F4A, and Navβ4 peptides can all accelerate fast inactivation (likely through open-channel block), can be quickly expelled from the channel pore upon repolarization, and have positive and hydrophobic residues that seem to be essential for inducing $I_{NaR}$ (*Dover et al., 2010*; *Lewis and Raman, 2011*; *Venkatesan et al., 2014*). These similarities lead us to propose that A-type FHFs induce $I_{NaR}$ via the N-terminus by a relief-of-open-channel-block mechanism, similar to the mechanism proposed for the Navβ4 peptide-dependent $I_{NaR}$ (*Lewis and Raman, 2014*).

Previous studies proposed that FHF4 isoforms may regulate Nav1.6 $I_{NaR}$ generation in Purkinje neurons (*Yan et al., 2014*; *White et al., 2019*). Although initially it was suggested that FHF4B indirectly enhanced $I_{NaR}$ by attenuating inactivation of Nav1.6 (*Yan et al., 2014*), a more recent study indicated that FHF4A directly mediated Nav1.6 $I_{NaR}$, showing a short peptide derived from a region of FHF4A adjacent to the β-trefoil core (residues 51–63) was able to induce robust $I_{NaR}$ in FHF4A knockout mice (*White et al., 2019*). However, our data do not support the idea that FHF4A alone can mediate Nav1.6 $I_{NaR}$ because FHF4A failed to mediate Nav1.6 $I_{NaR}$ in our heterologous expression system. Previously we demonstrated that FHF2A decreases Nav1.6 $I_{NaR}$ in DRG neurons. In contrast, FHF2B indirectly enhances $I_{NaR}$ generation in DRG neurons (*Barbosa et al., 2017*). FHF2B lacks the long N-terminus of A-type FHFs but retains the β-trefoil core found in all FHFs that can bind to the cytoplasmic tail of VGSCs and destabilizes Nav1.6 fast inactivation (also see *Figure 9h*). Interestingly, FHF4 knockout accelerates the onset of fast inactivation of sodium currents (*White et al., 2019*) and hyperpolarizes the voltage dependence of sodium current inactivation (*Bosch et al., 2015*) in cerebellar Purkinje neurons. This makes it possible that FHF4 isoforms regulate Nav1.6 $I_{NaR}$ in neurons, at least in part, by reducing channel fast inactivation, instead of directly mediating $I_{NaR}$.

While neither full-length Navβ4 nor full-length FHF4A has been shown to induce Nav1.6 $I_{NaR}$ in heterologous expression systems, our data show that FHF4A can induce long-term inactivation of Nav1.6 in ND7/23 cells (*Figure 10*), which is consistent with previous studies demonstrating that A-type FHFs induce long-term inactivation of Nav1.6 and Nav1.5 (*Dover et al., 2010*; *Yan et al., 2014*; *Yang et al., 2016*; *Barbosa et al., 2017*). One possible explanation for the induction of long-term inactivation versus the induction of $I_{NaR}$ generation is that A-type FHF N-terminus may bind more strongly to Nav1.6 and some other VGSCs than to Nav1.8 and Nav1.9. During repolarization, the driving force is only powerful enough to repel and unbind A-type FHF N-terminus from Nav1.8 and Nav1.9, but not from Nav1.6, and thus induces robust $I_{NaR}$ from Nav1.8 and Nav1.9, but long-term inactivation in Nav1.6. However, we cannot rule out the possibility that post-translational modifications of either Nav1.6 or FHF4A allows FHF4A to directly induce $I_{NaR}$ in Nav1.6, rather than inducing long-term inactivation, in specific neuronal populations such as cerebellar Purkinje neurons. ND7/23 cells are derived from the fusion of rat DRG neurons with the N18Tg2 mouse neuroblastoma cell lines, and thus may express proteins in addition to the transfected VGSCs and FHF that could be important for the differential effects on resurgent currents and long-term inactivation that we observed with Nav1.8 and Nav1.6.

Intriguingly, our data show that heterologous expression of A-type FHF is sufficient to induce $I_{NaR}$ in not only Nav1.8 and Nav1.9, but also, at least to some extent, in Nav1.5 and Nav1.7. This opens the door to further investigation of the molecular mechanism and the molecular manipulation of $I_{NaR}$. Our data show that F4A peptide at up to 1 mM, a nonphysiological concentration, is required to fully induce the same level of $I_{NaR}$ as those induced by full-length FHF4A. This concentration is 5-fold

to 10-fold higher than that of Navβ4 peptide (100–200 μM) that is required to reconstitute TTX-sensitive $I_{NaR}$. The unusually high concentration apparently is due to the lack of the FHF β-trefoil core. The β-trefoil core does not directly generate $I_{NaR}$, but it is likely crucial to facilitating $I_{NaR}$ generation induced by N-terminus residues in A-type FHFs, as the core binding to the cytoplasmic tail of VGSCs (*Liu et al., 2001*; *Goetz et al., 2009*) would greatly raise the local concentration of the N-terminus near the channel pore. In addition, our data and that of others demonstrate that the β-trefoil domain shifts the voltage dependence of steady-state inactivation in the positive direction, augmenting the 'window currents' region (see *Figure 1b and d*), where VGSCs activate but do not fully inactivate. *Lewis and Raman, 2013* showed that open-channel blockers might have higher affinity in VGSCs with DIVS4 deployed than with DIVS4 in the resting or partially deployed configuration. This also suggests that the molecular manipulation of $I_{NaR}$ might be achieved by inhibiting the interaction of A-type FHF β-trefoil core with VGSC C-terminus.

Overall, our work substantially increases understanding of the role of A-type FHFs in sensory neuron excitability. We show that A-type FHFs exert various impact on neuronal excitability by differentially modulating the activities of VGSC isoforms. The accumulation of long-term inactivation seems to be the predominant effect of A-type FHFs on TTX-sensitive VGSC isoforms, although FHF-mediated $I_{NaR}$ (only 0.3% of transient peak current) are inducible with some TTX-sensitive isoforms (*Figure 9c and f*). By promoting long-term inactivation, FHF2A accumulatively decreases TTX-sensitive sodium currents (e.g., Nav1.6, Nav1.7) by >20% (*Venkatesan et al., 2014*; *Effraim et al., 2019*). Prior studies demonstrated that A-type FHFs reduced action potential firing in hippocampal neurons, cerebellar granule neurons (*Dover et al., 2010*; *Venkatesan et al., 2014*), and medium-sized DRG neurons, where Nav1.6 channels are predominantly expressed (*Barbosa et al., 2017*). Nav1.8 and Nav1.9 are two TTX-resistant subtypes mainly expressed in nociceptive sensory neurons (*Fang et al., 2002*; *Cummins et al., 2007*). In *Figure 6*, we demonstrate that A-type FHF-mediated $I_{NaR}$ significantly upregulate excitability of nociceptive DRG neurons. The $I_{NaR}$ also result in broader action potentials and higher firing frequency. These observations are similar to those detected in Nav1.8 T790A-transfected DRG neurons, in which the T790A variant identified in the *Possum* transgenic mouse strain leads to increased $I_{NaR}$ and enhanced excitability (*Xiao et al., 2019*). In addition to DRG neurons, Nav1.8/Nav1.9 have been colocalized with A-type FHFs within other neuronal populations, such as trigeminal ganglion neurons, myenteric neurons, magnocellular neurosecretory cells of the supraoptic nucleus, the outer layers of the substantia gelatinosa, and cerebellar neurons in animal models of multiple sclerosis (*Craner et al., 2003*; *Vohra et al., 2006*; *Heanue and Pachnis, 2006*; *Huang et al., 2014*; *Osorio et al., 2014*). This opens up the possibility that A-type FHF-mediated $I_{NaR}$ extensively regulate excitability of the neurons throughout the PNS and CNS.

# Materials and methods

**Key resources table**

| Reagent type (species) or resource | Designation | Source or reference | Identifiers | Additional information |
|---|---|---|---|---|
| Strain, strain background (rat and male) | Sprague–Dawley | Envigo | | 7 weeks/~200 g |
| Cell line (mouse × rat hybridoma nerve) | ND7/23 cells | MilliporeSigma | CAT# 92090903 | |
| Cell line (*Homo sapiens*) | Hek293 cells | ATCC | CAT# CRL-1573 | |
| Cell line (*H. sapiens*) | Nav1.7 cells | Icagen LLC. | | |
| Cell line (*H. sapiens*) | Nav1.7_N945K cells | Icagen LLC. | | |
| Cell line (*H. sapiens*) | Nav1.9/β1/β2 cells | Icagen LLC (*Lin et al., 2016*) | | |
| Cell line (*H. sapiens*) | Nav1.9_K799N/β1/β2 cells | Icagen LLC (*Lin et al., 2016*) | | |
| Transfected construct (rat) | Nav1.8 shRNA | *Jarecki et al., 2010* | | pIRES-EGFP construct to transfect and express the shRNA |

*Continued on next page*

*Continued*

| Reagent type (species) or resource | Designation | Source or reference | Identifiers | Additional information |
|---|---|---|---|---|
| Transfected construct (rat) | Nav1.8 shRNA | *Jarecki et al., 2010* | | pIRES2-DsRed construct to transfect and express the shRNA |
| Transfected construct (rat) | FHF4 shRNA | *Wang et al., 2011a* | | Lentiviral construct to transfect and express the shRNA |
| Transfected construct (rat) | Scrambled shRNA | *Wang et al., 2011a* | | pAdTrack construct to transfect and express the shRNA |
| Antibody | FHF4 antibody (mouse monoclonal) | UC Davis/NIH NeuroMab Facility | Cat# N56/21 | IF (1:200) |
| Antibody | Anti-SCN4B antibody (rabbit polyclonal) | Abcam | Cat# ab80539 | IF (1:500) |
| Antibody | Anti-mouse IgG Alexa Fluor Plus 555 (goat polyclonal) | Invitrogen | Cat# A32727 | IF (1:1000) |
| Recombinant DNA reagent | pcDNA3.1-mouse Nav1.8 (plasmid) | GenScript (*Xiao et al., 2019*) | | |
| Recombinant DNA reagent | pcDNA3.1-human Nav1.8 (plasmid) | GenScript (*Xiao et al., 2019*) | | |
| Recombinant DNA reagent | FHF1A | Origene | CAT# RG215868 | Human tagged ORF clone: inserted into pCMV6-AC-GFP |
| Recombinant DNA reagent | FHF2A | GenScript (*Barbosa et al., 2015*) | | Inserted into pmTurquoise2-N1 |
| Recombinant DNA reagent | FHF2B | GenScript (*Barbosa et al., 2015*) | | Inserted into pmTurquoise2-N1 |
| Recombinant DNA reagent | FHF3A | Origene | CAT# RG207584 | Human tagged ORF clone: inserted into pCMV6-AC-GFP |
| Recombinant DNA reagent | FHF4A | Origene | CAT# RG219847 | Human tagged ORF clone: inserted into pCMV6-AC-GFP |
| Recombinant DNA reagent | Nav1.5 | *Xiao et al., 2019* | | Human ORF clone: inserted into pcDNA3.1 |
| Recombinant DNA reagent | Nav1.6 | GenScript | | Human ORF clone: inserted into pcDNA3.1 |
| Recombinant DNA reagent | Nav1.7 | *Xiao et al., 2019* | | Human ORF clone: inserted into pcDNA3.1-mod |
| Sequence-based reagent | Nav1.5 N927K_F | This paper | PCR primers | GGTCATTGGCAAGCTTGTGGTCCTGAATCTCTTCC |
| Sequence-based reagent | Nav1.5 N927K_R | This paper | PCR primers | GGAAGAGATTCAGGACCACAAGCTTGCCAATGACC |
| Peptide, recombinant protein | F2A | *Dover et al., 2010* | Amino acid sequence | AAAIASSLIRQKRQAREREK |
| Peptide, recombinant protein | 5Q | *Dover et al., 2010* | Amino acid sequence | AAAIASSLIRQQQQAQEQEQ |
| Peptide, recombinant protein | F4A | This paper | Amino acid sequence | AAAIASGLIRQKRQAREQHW |
| Peptide, recombinant protein | Navβ4 peptide | *Grieco et al., 2005* | Amino acid sequence | KKLITFILKKTREK |
| Commercial assay or kit | Site-directed mutagenesis | Stratagene | Cat# 200516 | |
| Commercial assay or kit | Lipofectamine 2000 | Invitrogen | Cat# 11668019 | |
| Chemical compound, drug | 5-Fluoro-2-deoxyuridine | Sigma-Aldrich | Cat# 856657 | |

*Continued on next page*

*Continued*

| Reagent type (species) or resource | Designation | Source or reference | Identifiers | Additional information |
|---|---|---|---|---|
| Chemical compound, drug | Uridine | Sigma-Aldrich | Cat# U3750 | |
| Chemical compound, drug | Tetrodotoxin (TTX) | Alomone Labs | Cat# T-550 | |
| Chemical compound, drug | Collagenase type 1 | Worthington Biochemical | Cat# LS004194 | |
| Chemical compound, drug | Neutral protease | Worthington Biochemical | Cat# LS02104 | |
| Software, algorithm | PulseFit | HEKA | | |
| Software, algorithm | PCLAMP | Molecular Devices | | |
| Software, algorithm | GraphPad Prism 5.0 | GraphPad Software | | |

## Plasmids, sodium channel constructs, and mutagenesis

Human FHF2A and FHF2B sequences were subcloned into pmTurquoise2-N1 vector as described by *Barbosa et al., 2017*. The pCMV6-AC-GFP plasmid encoding human FHF1A, FHF3A, or FHF4A was purchased from Origene USA Technologies, Inc (Rockville, MD). The cDNA construct encoding the human Nav1.5, mouse Nav1.8, and human Nav1.8 were subcloned into a pcDNA3.1 expression vector, respectively. The mutation N927K in Nav1.5 was constructed using the QuikChange XL (Stratagene) mutagenesis kit following the manufacturer's instructions (Stratagene). Mutations were confirmed by sequencing. The scrambled shRNA and FHF4shRNA constructs were generously provided by Dr. Geoffrey S Pitt (Duke University). The scrambled shRNA and FHF4shRNA were subcloned into pAdTrack and pLVTHM vectors, respectively.

## Cell culture and transfection

Rat DRG neurons were acutely dissociated and cultured according to the procedure described previously (*Xiao et al., 2019*). Briefly, young adult (8 weeks) Sprague–Dawley rats of either sex, in adherence to animal procedures approved by the Indiana University School of Medicine Institutional Animal Care and Use Committee, were killed by decapitation without anesthetization. All DRGs were removed quickly from the spinal cord and then incubated in Dulbecco's modified Eagle's medium (DMEM) containing collagenase (1 mg/ml) and protease (1 mg/ml). After the ganglia were triturated in DMEM supplemented with 10% fetal bovine serum (FBS), cells were seeded on glass coverslips coated with poly-D-lysine and laminin. Cultures were maintained at 37°C in a 5% $CO_2$ incubator. In order to be consistent with our previous studies (*Dib-Hajj et al., 2015*), the Helios Gene Gun (Bio-Rad Laboratories) was used to transiently cotransfect rat DRG neurons. Cells were cotransfected with an internal ribosome entry site–EGFP (IRES-EGFP) vector plasmid (or an IRES-DsRed vector plasmid) containing a Nav1.8 shRNA targeting the rat Nav1.8 but not the codon-optimized mouse Nav1.8 sequences. After transfection, DRG neurons were incubated in 10% FBS DMEM medium supplemented with mitotic inhibitors, 5-fluoro-2-deoxyuridine (50 μM, Sigma-Aldrich), and uridine (150 μM, Sigma-Aldrich), to prevent overgrowth of the supporting cells. DRG recordings were obtained from cells 2–5 days after transfection. Transfected cells were selected for recordings based on their ability to express EGFP. Under control conditions, the endogenous Nav1.8-type currents have an average current density of 947 ± 72 pA/pF (n = 70) and the Nav1.8 shRNA reduces endogenous Nav1.8-type current amplitudes in DRG neurons by 98% (*Xiao et al., 2019*).

Human Nav1.9, Nav1.9 K799N, Nav1.7, and Nav1.7 N945K channel cDNAs were stably expressed in the HEK-293-β1/β2 cell lines as described previously (*Lin et al., 2016*) and were provided by Icagen Inc (Durham, NC). They were incubated in 10% DMEM medium supplemented with G418 (400 mg/l) and puromycin (0.5 mg/l). Cell lines were transiently transfected by FHF1A, FHF2A, FHF2B, FHF3A, or FHF4A using the Invitrogen Lipofectamine 2000. Nav1.9 cells were seeded on glass coverslips and incubated at 30°C, 24 hr prior to patch-clamp recording.

HEK293 cells and ND7/23 cells were grown under standard tissue culture conditions (5% $CO_2$ and 37°C) in DMEM supplemented with 10% FBS. Using the Invitrogen Lipofectamine 2000, human Nav1.5

and the mutant construct (N927K) were transiently co-transfected with FHF2B, FHF2A, or FHF4A into HEK293 cells. The construct human Nav1.8 was transiently transfected into ND7/23 cells. The lipofect-amine-DNA mixture was added to the cell culture medium and left for 3 hr after which the cells were washed with fresh medium. Cells with green fluorescent protein fluorescence were selected for whole-cell patch-clamp recordings 36–72 hr after transfection. ND7/23 cells do not express endogenous Nav1.8 currents but TTX-sensitive sodium currents (*John et al., 2004*; *Lee et al., 2019*). Transfected ND7/23 cells were pretreated with 500 nM TTX to isolate Nav1.8 currents. No authentication of cell lines was performed. Mycoplasma infection was not detected when tested for.

### Electrophysiological recordings

Whole-cell voltage-clamp recordings were performed at room temperature (~21°C) using an EPC-10 amplifier and the Pulse program (HEKA Electronics). Recordings for hNav1.7 and hNav1.7 N945K were conducted at Icagen Inc under similar conditions but with an Axopatch 200B amplifier and PCLAMP software (Molecular Devices).

For voltage-clamp recordings, fire-polished electrodes (1.0–2.0 MΩ) were fabricated from 1.7 mm capillary glass using a P-97 puller (Sutter Instruments), and the tips were coated with sticky wax (KerrLab) to reduce electrode capacitance and enable increased series resistance compensation. The pipette solution contained (in mM) 140 CsF, 1.1 EGTA, 10 NaCl, and 10 HEPES, pH 7.3. The bathing solution was (in mM) 130 mM NaCl, 30 mM TEA chloride, 1 mM $MgCl_2$, 3 mM KCl, 1 mM $CaCl_2$, 0.05 mM $CdCl_2$, 10 mM HEPES, and 10 mM D-glucose, pH 7.3 (adjusted with NaOH). TTX (500 nM) was added to the bath solution in order to block endogenous TTX-sensitive currents in DRG neurons, Nav1.9 and K799N stable cells, and cells expressing Nav1.8, Nav1.5, and the mutant N927K. The liquid junction potential for these solutions was <8 mV; data were not corrected to account for this offset. The offset potential was zeroed before contacting the cell. After establishing the whole-cell recording configuration, the resting potential was held at –120 mV or –100 mV for 5 min to allow adequate equilibration between the micropipette solution and the cell interior. Linear leak subtraction, based on resistance estimates from 4 to 5 hyperpolarizing pulses applied before the depolarizing test potential, was used for all voltage-clamp recordings. Membrane currents were usually filtered at 5 kHz and sampled at 20 kHz. Voltage errors were minimized using 70–90% series resistance compensation, and the capacitance artifact was canceled using the computer-controlled circuitry of the patch-clamp amplifier.

### Steady-state activation

Families of sodium currents were induced by 50 ms depolarizing steps to various potentials ranging from –120 to +40 mV in 5 mV (or 10 mV) increments. The conductance was calculated using the equation G(Nav) = I/(V - Vrev) in which I, V, and Vrev represent inward current value, membrane potential, and reversal potential, respectively.

### Steady-state inactivation

The voltage dependence of steady-state inactivation was estimated using a standard double-pulse protocol in which sodium currents were induced by a 50 ms depolarizing potential of 0 mV following a 500 ms prepulse at potentials that ranged from –130 to +10 mV with a 10 mV increment. Currents were plotted as a fraction of the maximum peak current. To obtain the midpoint voltages ($V_{1/2}$) and slope factor (k), the curves of both steady-state activation and inactivation were fitted to a Boltzmann function.

### Recovery from inactivation

Recovery from inactivation was assayed by the protocol that the cells were prepulsed to 0 mV for 50 ms to inactivate sodium channels and then brought back to –100 mV for increasing recovery durations before the test pulse to 0 mV.

### Resurgent currents

$I_{NaR}$ were elicited by repolarizing voltage steps from +10 mV to −100 for 100 ms (200 ms, or 1000 ms as indicated in *Figures 2–9* [inset]) in –5 mV increments, following a 20 ms depolarizing potential of +30 mV (or +100 mV). To avoid contamination from tail currents, Navβ4-induced Nav1.5 $I_{NaR}$ were

measured after 3.0 ms into the repolarization pulse, FHF-induced Nav1.5, Nav1.7, and Nav1.9 $I_{NaR}$ were measured after 4.0 ms into the repolarization pulse, and FHF-induced Nav1.8 $I_{NaR}$ were measured after 20 ms into the repolarization pulse. The relative $I_{NaR}$ in Nav1.5, Nav1.7, and Nav1.8 were calculated by normalizing to the peak transient current elicited at 0 mV, but the relative Nav1.9 resurgent currents were calculated by normalizing to the peak transient current at –30 mV.

For current-clamp recordings, fire-polished electrodes (4.0–5.0 MΩ) were fabricated from 1.2 mm capillary glass using a P-97 (Sutter Instruments). The pipette solution contained the following (in mM): 140 KCl, 5 MgCl$_2$, 5 EGTA, 2.5 CaCl$_2$, 4 ATP, 0.3 GTP, and 10 HEPES, pH 7.3 (adjusted with KOH). The bathing solution contained the following (in mM): 140 NaCl, 1 MgCl2, 5 KCl, 2 CaCl$_2$, 10 HEPES, and 10 glucose, pH 7.3 (adjusted with NaOH). Neurons were allowed to stabilize for 3 min in the current-clamp mode before initiating current injections to measure action potential activity.

## Immunocytochemistry

Immunocytochemistry was performed according to the procedure as described previously (*Liu et al., 2001*). Briefly, the Helios Gene Gun (Bio-Rad Laboratories) was used to transiently transfect the scrambled shRNA, or FHF4shRNA in cultured DRG neurons. Three days after transfection, DRG neurons were fixed with 4% PFA (0.1 M phosphate buffer, pH 7.4) for 20 min and washed in PBS. Cells were then permeabilized in 1% Triton X-100 in PBS for 20 min at room temperature (~21°C), washed in PBS, blocked for 2 hr (10% normal goat serum, 0.1% Triton X-100 in PBS) at room temperature, and washed with PBS. Cells were then incubated with monoclonal FHF4 antibody (1:200, N56/21, UC Davis/NIH NeuroMab Facility) or polyclonal anti-Navβ4 antibody (1:500, #Ab80539, Abcam) diluted in blocking solution overnight at 4°C. After additional PBS washes, cells were incubated with secondary antibody Alexa Fluor Plus 555 Goat Anti-Mouse IgG (Invitrogen) in blocking solution at 1:1000 concentration for 2 hr at room temperature. Coverslips were mounted in Prolong Gold Antifade (Invitrogen) and DRG neurons imaged using Leica Microscope system with a ×20 objective (Biocompare). Images were analyzed with Leica software, and corrected mean cell fluorescence was calculated in Excel (Microsoft) by applying measurements obtained from image analysis using the equations: CMCF = (mean fluorescence intensity of selected cell) – (mean fluorescence of background).

## Experimental design and statistical analysis

The acquisition of control and experimental data was randomized. Data were analyzed using the software programs PulseFit (HEKA) and GraphPad Prism 5.0 (GraphPad Software, Inc, San Diego, CA). All data are shown as mean ± SE. The number of separate experimental cells is presented as *n*. Statistical analysis was performed by Student's *t*-test, one-way ANOVA and $\chi^2$ analysis, and p<0.05 indicates a significant difference.

# Acknowledgements

This work was supported, in whole or in part, by the National Institute of Neurological Disorders and Stroke of the National Institutes of Health under Award Numbers R21NS109896 (to YX and TRC) and NS053422 (to TRC), and the Indiana Spinal Cord & Brain Injury Research Fund from the Indiana State Department of Health (2020) (to YX). We thank Dr. Geoffrey S Pitt for generously providing the scrambled shRNA and FHF4shRNA constructs.

# Additional information

### Competing interests
Jonathan W Theile, Zhixin Lin: is affiliated with Icagen, LLC. The author has no financial interests to declare. The other authors declare that no competing interests exist.

## Funding

| Funder | Grant reference number | Author |
|---|---|---|
| National Institute of Neurological Disorders and Stroke | NS109896 | Yucheng Xiao<br>Theodore R Cummins |
| National Institute of Neurological Disorders and Stroke | NS053422 | Theodore R Cummins |
| Indiana Spinal Cord & Brain Injury Research Fund from the Indiana State Department of Health | 2020 | Yucheng Xiao |

The funders had no role in study design, data collection and interpretation, or the decision to submit the work for publication.

## Author contributions

Yucheng Xiao, Conceptualization, Data curation, Formal analysis, Funding acquisition, Investigation, Project administration, Writing - original draft, Writing – review and editing; Jonathan W Theile, Conceptualization, Investigation, Writing – review and editing; Agnes Zybura, Yanling Pan, Formal analysis, Investigation, Writing – review and editing; Zhixin Lin, Investigation; Theodore R Cummins, Conceptualization, Funding acquisition, Project administration, Resources, Supervision, Writing - original draft, Writing – review and editing

## Author ORCIDs

Yucheng Xiao http://orcid.org/0000-0002-0298-7158
Theodore R Cummins http://orcid.org/0000-0001-9509-6380

## Ethics

This study was performed in strict accordance with the recommendations in the Guide for the Care and Use of Laboratory Animals of the National Institutes of Health. All of the animals were handled according to approved institutional animal care and use committee (IACUC) protocols (#SC307R) of the Indiana University - Purdue University Indianapolis.

## Decision letter and Author response

Decision letter https://doi.org/10.7554/eLife.77558.sa1
Author response https://doi.org/10.7554/eLife.77558.sa2

# Additional files

## Supplementary files

• Transparent reporting form

## Data availability

All data generated or analyzed during this study are included in the manuscript.

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
