## [Editor Report]

This is an exciting and important study that constitutes a major advance in the molecular understanding of resurgent Na current. Reproducing resurgent current by expression of two proteins has never been done: here, the authors have for the first time molecularly reconstituted Na channels that produce resurgent Na current. Not only do these experiments satisfactorily and convincingly address a long-standing question in the field, but they also open the door to molecular manipulation of this current, potentially of significant practical use given the proposed role of the current in several disorders and disease states, including pain. The work will be of interest to many neuroscientists.

---

## [Decision Letter]

**Decision letter after peer review:**

Thank you for submitting your article "A-type FHFs mediate resurgent currents through TTX-resistant voltage-gated sodium channels" for consideration by *eLife*. Your article has been reviewed by 3 peer reviewers, and the evaluation has been overseen by a Reviewing Editor and Kenton Swartz as the Senior Editor. The following individuals involved in the review of your submission have agreed to reveal their identity: Bruce P. Bean (Reviewer #1); Mitchell Goldfarb (Reviewer #2).

Please see the individual reviews as they offer feedback for the authors and suggestions for revision.

*Reviewer #1 (Recommendations for the authors):*

Figure leg 2 and 3: (c,g) "Voltage dependence of the relative Nav1.8 and Nav1.9 INaR mediated by FHF1A – FHF4A. " It is not immediately obvious what "relative" means here, and it should be specified more exactly. From the Methods, I believe it means relative to peak transient current and this should be stated. The labeling of the y-axes was a little confusing because at first I assumed "Ip" might refer to persistent current. Using "It" for transient current seems more conventional and might be more quickly understandable.

Line 151. "The relative amplitudes were 1.5% {plus minus} 0.1% and 2.5% {plus minus} 0.4% in Nav1.8 at -15 mV…" It is not stated percent of what…presumably peak transient current.

Line 261. "Furthermore, because changes at this position did alter A-type FHF mediated INaR …" Should be "did not alter".

Line434. "tips were coated with sticky wax (Kerr Lab) to minimize capacitive artifacts and increase series resistance compensation.:. Would be clearer as to "enable increased series resistance compensation".

Line 468. The sentence is missing an "INaR".

*Reviewer #2 (Recommendations for the authors):*

This paper is strong in its present form. I do not have criticisms of the data or narrative.

*Reviewer #3 (Recommendations for the authors):*

The authors are to be congratulated on a beautiful study.

1. Intro lines 47 and 48. "classic sodium currents that are activated during the depolarizing phase of action potentials, INaR are atypical sodium currents evoked during the repolarizing phase." It would be more correct to say "classic sodium currents that are activated by step depolarizations, INaR are atypical sodium currents evoked by step repolarizations." (Strictly speaking, during spiking, there is no way to say that an open channel is passing a resurgent or a transient current. Transient and resurgent currents are only evoked under voltage clamp, when (most) channels are preconditioned into a common state and then made to change state with a voltage step. The best one could say might be that transient current comes from the closed to open transition and resurgent current from a blocked to open transition.)

2. It would be helpful somewhere to relate the FHF terminology to the FGF terminology (FHF4A = FGF14-1a, etc.) for readers who don't immediately know the two nomenclatures.

3. Results line 99, please indicate in the text or legend the voltage of recovery from inactivation. (It is only in the methods.)

4. Results line 129, "recovery from inactivation of NaV1.9 persistent currents." It seems that the word "persistent" doesn't belong here and could simply be omitted. (If they are persistent, why do they need to recover from inactivation?)

5. Results line 179, "transfected with a scrambled shRNA" please indicate that this is scrambled FHF4A shRNA.

6. Results line 186, Figure 5f, could it be resolved whether the shRNA to FHF4A slowed or sped the decay of transient current evoked by a step depolarization?

7. Figure 5-figures supplement 1. Suggestion: indicate on the figure that the measurements are related to NaVbeta4.

8. Results line 204 "current threshold of action potential firing." Is "rheobase" what is meant?

9. Figure 3c, 3f, and related, please indicate in the text or legend that the y-axis is resurgent normalized to transient. The use of "Ip" for transient current made it seem that the normalization was too persistent a current or that there was a typo.

10. Table 1. Please indicate that the voltages are not corrected for the 8-mV junction potential for those readers interested in absolute voltages.

11. Discussion line 326. "do not support the idea that FHF4A serves as a direct mediator of NaV1.6 INaR, because FHF4A failed to mediate NaV1.6 INaR in our heterologous expression system." The authors may well be right in the conclusion, but not for the reason stated. NaVbeta4 also fails to mediate INaR in heterologous systems, but its knockdown, including by these authors, suggests a role. It may be the heterologous systems that complicate matters. Perhaps consider modifying the text to something like "do not support the idea that FHF4A alone can mediate NaV1.6 INaR, because FHF4A failed to mediate NaV1.6 INaR in our heterologous expression system."

12. Discussion line 336 "blocking particle independent mechanisms" The key states responsible for resurgent current in this computational model function as open channel blocked states but just are not labeled as such and draw a conclusion that is not supported by the data even within that paper. It is risky to cite this as evidence for non-block mechanisms. If models are brought up at all, it might be more prudent to state that different models can mimic resurgent currents and include Raman and Bean 2001, which is more mechanistic, unless these authors are convinced that the later model supplants the earlier mechanism.

---

## [Author Response]

Reviewer #1 (Recommendations for the authors):Figure leg 2 and 3: (c,g) "Voltage dependence of the relative Nav1.8 and Nav1.9 INaR mediated by FHF1A – FHF4A. " It is not immediately obvious what "relative" means here, and it should be specified more exactly. From the Methods, I believe it means relative to peak transient current and this should be stated. The labeling of the y-axes was a little confusing because at first I assumed "Ip" might refer to persistent current. Using "It" for transient current seems more conventional and might be more quickly understandable.

We apologize for the confusions. According to the reviewer’s suggestions, we have indicated that “Nav1.8 and Nav1.9 I_NaR_ are normalized to the peak transient currents elicited at 0 mV and -30 mV, respectively” in Figure 2c,g legend (please see line 710) and that “Nav1.8 I_NaR_ are normalized to the peak transient current elicited at 0 mV” in Figure 3c legend (please see line 729). We have changed the labeling of the y-axes “Ip” to “It”. The same change has also been made in other figures (Figures 3-9).

Line 151. "The relative amplitudes were 1.5% {plus minus} 0.1% and 2.5% {plus minus} 0.4% in Nav1.8 at -15 mV…" It is not stated percent of what…presumably peak transient current.

We have indicated that the relative amplitudes were 1.5% ± 0.1% and 2.5% ± 0.4% of peak transient current” (please see line 156). Thank you for requesting that clarification.

Line 261. "Furthermore, because changes at this position did alter A-type FHF mediated INaR …" Should be "did not alter".

We thank the reviewer for the careful reading of the manuscript. The error has been corrected.

Line434. "tips were coated with sticky wax (Kerr Lab) to minimize capacitive artifacts and increase series resistance compensation.:. Would be clearer as to "enable increased series resistance compensation".

As suggested by the reviewer, we have changed the text “…to minimize capacitive artifacts and increase series resistance compensation” to “… to reduce electrode capacitance and enable increased series resistance compensation” (please see line 445).

Line 468. The sentence is missing an "INaR".

We have added the missing word “I_NaR_” (please see line 472).

Reviewer #2 (Recommendations for the authors):This paper is strong in its present form. I do not have criticisms of the data or narrative.Reviewer #3 (Recommendations for the authors):The authors are to be congratulated on a beautiful study.1. Intro lines 47 and 48. "classic sodium currents that are activated during the depolarizing phase of action potentials, INaR are atypical sodium currents evoked during the repolarizing phase." It would be more correct to say "classic sodium currents that are activated by step depolarizations, INaR are atypical sodium currents evoked by step repolarizations." (Strictly speaking, during spiking, there is no way to say that an open channel is passing a resurgent or a transient current. Transient and resurgent currents are only evoked under voltage clamp, when (most) channels are preconditioned into a common state and then made to change state with a voltage step. The best one could say might be that transient current comes from the closed to open transition and resurgent current from a blocked to open transition.)

We agree with the reviewer that the statement needed to be improved. We have changed the text (“Unlike classic sodium currents that are activated by step depolarizations, I_NaR_ are atypical sodium currents evoked by step repolarizations”) as suggested by the reviewer (see lines 47-48).

2. It would be helpful somewhere to relate the FHF terminology to the FGF terminology (FHF4A = FGF14-1a, etc.) for readers who don't immediately know the two nomenclatures.

The FGF terminology “FHF1A (or FGF12-1a), FHF2A (or FGF13-1a), FHF3A (or FGF11-1a), FHF4A (or FGF14-1a)” has been indicated (please see lines 63-64). “FHF2B (also known as FGF13-1b)” has been indicated, too (please see line 65).

3. Results line 99, please indicate in the text or legend the voltage of recovery from inactivation. (It is only in the methods.)

The voltage of recovery from inactivation is now indicated in Figure 1 legend of (please see line 707).

4. Results line 129, "recovery from inactivation of NaV1.9 persistent currents." It seems that the word "persistent" doesn't belong here and could simply be omitted. (If they are persistent, why do they need to recover from inactivation?)

As suggested by the reviewer, the word “persistent” is now simply omitted.

5. Results line 179, "transfected with a scrambled shRNA" please indicate that this is scrambled FHF4A shRNA.

“scrambled FHF4A shRNA” is now indicated. Thank you for requesting this clarification.

6. Results line 186, Figure 5f, Could it be resolved whether the shRNA to FHF4A slowed or sped the decay of transient current evoked by a step depolarization?

We measured the time constants of the decay of transient current evoked at +30 mV. The values of the τ_fast_ component are 1.51 ± 0.13 ms (scramble) and 1.57 ± 0.14 ms (FHF4shRNA), respectively. The values of the τ_slow_ component are 11.93 ± 1.25 ms (scramble) and 13.55 ± 1.10 ms (FHFshRNA), respectively. FHF4 knockdown shows a slight tendency to slow the decay of transient current; however, the change in both τ_fast_ and τ_slow_ is not statistically significant (p > 0.05). A-type FHFs have the same binding site at sodium channel C-tails. One possible explanation is that after FHF4A knockdown, other A-type FHF isoforms (e.g.FHF2A) may take its binding site. Their N-terminus, like FHF4A N-terminus, induces open-channel block and speeds the decay of transient current so that FHF4A knockdown does not change the decay significantly.

7. Figure 5-figures supplement 1. Suggestion: indicate on the figure that the measurements are related to NaVbeta4.

As suggested by the reviewer, “Navβ4” has been indicated on the labeling of y-axes in Figure 5-figures supplement 1.

8. Results line 204 "current threshold of action potential firing." Is "rheobase" what is meant?

We agree that “rheobase” is more accurate than “current threshold”. We have changed “current threshold” to “rheobase” (please see lines 209, 225, 778 and Figure 6c).

9. Figure 3c, 3f, and related, please indicate in the text or legend that the y-axis is resurgent normalized to transient. The use of "Ip" for transient current made it seem that the normalization was too persistent a current or that there was a typo.

We agree that the use of “Ip” may confuse the readers. We have changed “Ip” to “It” in all figures.

10. Table 1. Please indicate that the voltages are not corrected for the 8-mV junction potential for those readers interested in absolute voltages.

The statement has been indicated in table 1.

11. Discussion line 326. "do not support the idea that FHF4A serves as a direct mediator of NaV1.6 INaR, because FHF4A failed to mediate NaV1.6 INaR in our heterologous expression system." The authors may well be right in the conclusion, but not for the reason stated. NaVbeta4 also fails to mediate INaR in heterologous systems, but its knockdown, including by these authors, suggests a role. It may be the heterologous systems that complicate matters. Perhaps consider modifying the text to something like "do not support the idea that FHF4A alone can mediate NaV1.6 INaR, because FHF4A failed to mediate NaV1.6 INaR in our heterologous expression system."

We thank the reviewer for the important suggestion. We have modified the text as suggested by the reviewer (please see line 331).

12. Discussion line 336 "blocking particle independent mechanisms" The key states responsible for resurgent current in this computational model function as open channel blocked states but just are not labeled as such and draw a conclusion that is not supported by the data even within that paper. It is risky to cite this as evidence for non-block mechanisms. If models are brought up at all, it might be more prudent to state that different models can mimic resurgent currents and include Raman and Bean 2001, which is more mechanistic, unless these authors are convinced that the later model supplants the earlier mechanism.

We agree with the reviewer that the “blocking particle independent mechanisms” is controversial. To avoid the confusion, we have removed the discussion of this computational modeling study.